# DMEL: SPEECH TOKENIZATION MADE SIMPLE

## ABSTRACT

Large language models have revolutionized natural language processing by leveraging self-supervised pretraining on vast textual data. Inspired by this success, researchers have investigated complicated speech tokenization methods to discretize continuous speech signals so that language modeling techniques can be applied to speech data. However, existing approaches either model semantic (content) tokens, potentially losing acoustic information, or model acoustic tokens, risking the loss of semantic (content) information. Having multiple token types also complicates the architecture and requires additional pretraining. Here we show that discretizing mel-filterbank channels into discrete intensity bins produces a simple representation (`dMel`), that performs better than other existing speech tokenization methods. Using an LM-style transformer architecture for speech-text modeling, we comprehensively evaluate different speech tokenization methods on speech recognition (ASR) and speech synthesis (TTS). Our results demonstrate the effectiveness of `dMel` in achieving high performance on both tasks within a unified framework, paving the way for efficient and effective joint modeling of speech and text. The code will be open sourced.

## 1 INTRODUCTION

Large language models (LLMs) have achieved remarkable success in various natural language processing tasks by leveraging self-supervised pretraining on massive amounts of textual data (Brown et al., 2020). Inspired by this success, numerous works (Borsos et al., 2023; Rubenstein et al., 2023; Zhang et al., 2023a; Wang et al., 2023a) have sought to extend the language modeling approach to speech processing, aiming to build unified models capable of both speech understanding and generation tasks. However, a key challenge lies in the continuous nature of speech signals, necessitating effective tokenization methods to discretize the input for language model-based processing.

Current speech tokenization approaches can be broadly categorized into two types: semantic (content) tokens and acoustic tokens[1]. Semantic tokens, extracted from self-supervised (SSL) pretrained speech models (Baevski et al., 2020; Hsu et al., 2021), where the speech signal is first encoded into speech representations and then clustered into semantic tokens with $k$-means method. However, such SSL pretrained models are not useful for high fidelity speech synthesis as speaker identity and other details of raw speech are lost in training (Borsos et al., 2023). Conversely, acoustic tokens can be obtained from audio compression models that are trained to compress the speech signal into codebook indices with residual vector quantization (RVQ) and reconstruction objectives (Zeghidour et al., 2021; Défossez et al., 2022). These tokens prioritize acoustic reconstruction but lose semantic information which can lead to poorer results in generating audio (Wang et al., 2023a).

To combine the advantages of both semantic and acoustic tokens, AudioLM (Borsos et al., 2023) proposed to model both semantic tokens and acoustic tokens with 3 stages: semantic modeling, coarse acoustic modeling, and fine acoustic modeling. The coarse-to-fine modeling strategy is designed to match the residual structure of RVQ based acoustic tokens. This solution addresses both content and speech quality, but its multi-stage hierarchical structure complicates the model and can lead to slower training and inference. Another solution is to combine the semantic and acoustic features together. Zhang et al. (2023b) proposed to distill the semantic tokens into the acoustic token's first residual channel during the training of the RVQ model in a teacher-student manner. In this way, the new feature can preserve the semantic information better and also reconstruct high quality speech signals.

---

[1]We use a word 'semantic' with the meaning of 'content' to keep prior work notation (Borsos et al., 2023).

In this paper, we raise the following fundamental question – ***do we really need to separate speech into semantic and acoustic tokens first, and process them with idiosyncratic architectures?*** We propose a simple alternative called `dMel` (see Figure 1) that discretizes log mel-filterbanks (Mel) energies directly into ordinal bins. Intriguingly, we find that discretizing Mel has little impact on the ability of off-the-shelf Mel vocoders to reconstruct waveforms[2]. In Table 1 we show different vocoders to reconstruct waveforms from Mel and discretized Mel (`dMel`) computed on them, as well as ASR models trained on Mel and `dMel`.

We find that the word error rate (WER) of an ASR system run on the reconstructed waveforms, is quite similar to the WER of the same system run on the ground-truth audio, showing that `dMel` captures the acoustic information needed to reconstruct good waveforms. Similarly, we find that the WER of ASR models trained on Mel and `dMel` are similar, indicating that `dMel` are good at preserving semantic content that can easily be found by ASR models. This shows that discretizing Mel has limited impact on information content.

Table 1: Impact of discretization.

| | Reconstruction WER (%) | | Recognition WER (%) | |
|---|---|---|---|---|
| | P-WaveGAN[1] | HifiGAN[2] | Seq2seq[3] | CTC[4] |
| Ground-truth | 2.02 | | - | |
| Mel | 2.13 | 2.08 | 2.4 | 2.1 |
| dMel | 2.23 | 2.11 | 2.5 | 2.1 |

[*] Yamamoto et al. (2020)[1], Kong et al. (2020)[2], Dong et al. (2018)[3], Graves et al. (2006)[4] Configurations are detailed in Sec. 3.

By operating on the log mel-filterbanks and preserving the frequency and intensity information (with some loss of resolution from discretization), `dMel` *inherently preserves both semantic and acoustic information in a unified representation*, without the need for separate tokenization or additional pretraining of a tokenization model. There are many advantages to discretizing log mel-filterbanks:

- Log mel-filterbanks is an interpretable representation of speech signal, where both the semantic and acoustic information is preserved. As discretization has little impact, `dMel` inherits the properties.
- `dMel` is a *model-free representation grounded in raw acoustic space*. As a result it can be converted to waveforms by any mel-filterbank vocoder, unlike other tokenization schemes that have feature representations that are intricately coupled to both the encoder and the decoder.
- Different channels of `dMel` do not have the complex hierarchical dependencies on each other that is typical of coarse-to-fine acoustic tokens; we find that they can be ***modeled independently*** in each frame using a simple decoder-only (LM-style) transformer architecture.

While data-driven methods can be improved by introducing more diverse data, they inherently suffer from information loss due to their neural compression nature – they learn to discard information based on training data, which may prove crucial in unseen conditions. In contrast, mel-spectrogram is a physics-based signal representation that:

- Preserves frequency components through principled transformation – while it does not retain phase information, this aligns with human auditory perception primarily sensitive to magnitude spectrum.
- Has demonstrated robust performance over decades of speech processing research.
- Does not require data-dependent training to handle different acoustic conditions.
- Maintains consistent behavior across various noise conditions due to its deterministic nature.

Through comprehensive evaluations, we show that using `dMel` allows us to use a single decoder-only model, and achieve high performance on both automatic speech recognition (ASR) and text-to-speech (TTS) tasks. The ASR task validates that `dMel` preserves semantic information, while the TTS task shows that `dMel` are useful for high-fidelity acoustic reconstruction of speech. We also compare `dMel` to other tokenization methods and find that `dMel` achieves the best WER for ASR task, which indicates that the semantic information is well preserved. Also, `dMel` achieves the lower WER score for TTS task when using WhisperX (Bain et al., 2023) for automatic evaluation and we find model trained with `dMel` can generate long and natural speech samples (see Supplementary Material).

## 2 METHOD

In this section, we first introduce our proposed `dMel` speech tokenization method, which discretizes log mel-filterbanks energies directly into bins. We then describe our unified LM-style transformer model for ASR and TTS tasks, which leverages `dMel` for speech tokenization. The model architecture is illustrated in Figure 2.

---

[2]We used vocoders from `https://github.com/kan-bayashi/ParallelWaveGAN`.

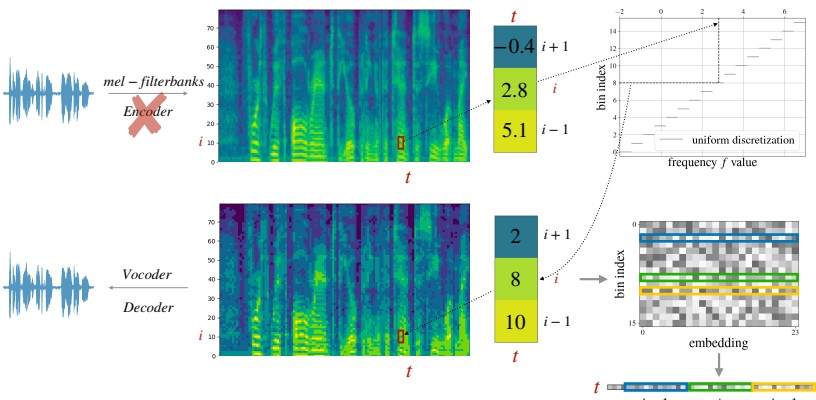

Figure 1: Prior works on speech tokenization use either heavy self-supervised pretrained encoders (Baevski et al., 2020; Hsu et al., 2021) to extract semantic tokens (and train a separate decoder for it (Lakhotia et al., 2021)) or learn compression encoder-decoder models with residual vector quantizations (Zeghidour et al., 2021; Défossez et al., 2022) to obtain acoustic tokens. By contrast we eliminate the encoder and simply discretize mel-filerbanks (dMel) to encode audio, and use a simple mel-filterbank vocoder (Yamamoto et al., 2020) to reconstruct speech signals.

## 2.1 DMEL SPEECH TOKENIZER

Different from existing VQ-VAE (Borsos et al., 2023; Zhang et al., 2023b; Kim et al., 2024; Zeghidour et al., 2021) based speech tokenizers, we propose a discretized log mel-filterbanks based speech tokenizer. The outline of the discretization method is shown in Figure 1. Later in the paper, we show that this tokenizer allows the model to process the input speech signal efficiently and capture the relevant acoustic features for both ASR and TTS tasks.

We denote tensors as $\mathbf{X}$ while $\mathbf{X}_{i,...}$ denote the $(i,...)$-th component of tensor $\mathbf{X}$. First, the speech tokenizer takes the input speech signal $\mathbf{x}$ and computes the log mel-filterbanks representation $\mathbf{M}$:

$$\mathbf{M} = \text{Mel}(\mathbf{x}), \tag{1}$$

where $\text{Mel}(\cdot)$ represents the function that computes the log mel-filterbanks, $\mathbf{M} \in \mathbb{R}^{T \times N}$, $N$ is the number of log mel-filterbanks and $T$ is the number of frames in the spectrogram.

**Tokenization** To discretize the log mel-filterbanks representation $\mathbf{M}$ into speech tokens, we adopt a codebook $\mathbf{C}$. In this paper, we apply a simple linear discretization, so that the codebook $\mathbf{C} \in \mathbb{R}^{2^K}$ and its values are evenly spaced in the range of the log mel-filterbanks values:

$$m = \min_{t,i}(\mathbf{M}_{t,i}), \qquad M = \max_{t,i}(\mathbf{M}_{t,i}) \qquad \delta = \frac{M - m}{2^K}, \tag{2}$$

$$\mathbf{C} = \left[ m, \, m + \delta, \, m + 2\delta, \, \ldots, \, m + (2^K - 1)\delta \right]. \tag{3}$$

In practice, we compute the minimum $m$ and maximum $M$ values of log mel-filterbanks across the entire dataset to define the codebook $\mathbf{C}$. Then we map a magnitude $\mathbf{M}_{t,i}$ of every frequency channel $i = 1 \ldots N$ for the time frame $t = 1 \ldots T$ into a bin index of the codebook $\mathbf{C}$ in the following way:

$$\mathbf{S}_{t,i} = \text{Discretize}(\mathbf{M}_{t,i}) = \text{argmin}_j |\mathbf{M}_{t,i} - \mathbf{C}_j| \tag{4}$$

where $\mathbf{S} \in \mathbf{B}^{T \times N}$ represents the discretized log mel-filterbanks (dMel) with $\mathbf{B} = \{j | j = 1, 2, 3, \ldots 2^K\}$ and $\mathbf{S}_t \in \mathbf{B}^N$ being the $t$-th speech token. As the codebook $\mathbf{C}$ has $2^K$ distinct values and thus number of bins $|\mathbf{B}| = 2^K$, each speech token is represented by $N \cdot K$ bits where every $K$ bits are used to represent one of $N$ frequency channels.

**Detokenization** To reconstruct the speech signal $\mathbf{x}$ from the speech tokens $\mathbf{S}$, we first transform bin indices back to the log mel-filterbanks representation via the codebook $\mathbf{C}$:

$$\hat{\mathbf{M}}_{t,i} = \mathbf{C}_{\mathbf{S}_{t,i}}. \tag{5}$$

Then, we apply a vocoder (Yamamoto et al., 2020) to transform reconstructed log mel-filterbanks $\hat{\mathbf{M}}_{t,i}$ back into the time domain signal $\mathbf{x}$. The vocoder is trained independently and is not part of the transformer decoder-based model.

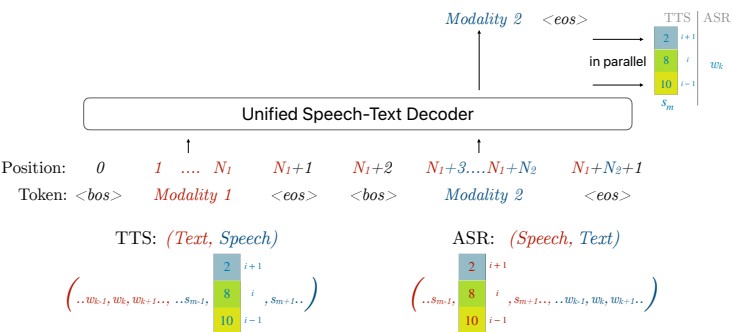

Figure 2: Unified Speech-Text Transformer Decoder with speech tokens as dMel.

## 2.2 UNIFIED SPEECH-TEXT TRANSFORMER DECODER

Modeling speech and text sequences jointly is essential for a model to understand and generate both modalities. However, it is challenging to design a unified model that can handle both speech-to-text and text-to-speech effectively. In this work, we apply a unified LM-style transformer model that takes speech and text tokens as input and generates the output tokens in the target sequence. The model is trained in end-to-end on a combined dataset of speech and text pairs, enabling it to learn the joint representations for ASR and TTS tasks. As we show in the rest of the paper, the crucial part for the joint model training is the proper speech tokenization which dMel provides.

**Token Representation** For text data, we apply a character-level tokenizer to convert the input text into a sequence of text tokens. The text tokens are passed through an embedding layer, Embed$(\cdot)$ : $\{j|j = 1, 2, 3 \ldots L\} \to \mathbb{R}^D$, where $D$ is the embedding dimension and $L$ is the vocabulary size. The dimension of the speech token embedding is set to be the same as the text token embedding $D$ and no further mapping is required. The motivation for using a character-level tokenizer is to reduce the vocabulary size $L$ and improve the model's generalization ability. Also, character tokens can capture the fine-grained linguistic features that are essential for both ASR and TTS tasks.

For speech signal, we apply the dMel speech tokenizer to convert the input speech signal into a sequence of speech tokens. Then, the speech tokens $\mathbf{S} \in \mathbf{B}^{T \times N}$ are passed through a learnable embedding layer, Embed$(\cdot) : \mathbf{B} \to \mathbb{R}^d$, and a learnable linear layer, Linear$(\cdot) : \mathbb{R}^{N \times d} \to \mathbb{R}^D$, to obtain the speech token representation $\mathbf{E} \in \mathbb{R}^{T \times D}$:

$$\mathbf{E}_t = \text{Linear}(\mathbf{E}'_t), \text{ and } \mathbf{E}'_t = \text{Concatenate}([\text{Embed}(\mathbf{S}_{t,1}), \text{Embed}(\mathbf{S}_{t,2}), \ldots, \text{Embed}(\mathbf{S}_{t,N})]), \quad (6)$$

where $\mathbf{E}_t \in \mathbb{R}^D$ is the speech token representation. Here, for every time frame $t$, a speech token $\mathbf{S}_t$ is processed *in parallel and independently* for every frequency channel $i$ by Embed$(\mathbf{S}_{t,i})$ mapping, and then embeddings of all frequency channels are stacked together to form one vector representation $\mathbf{E}'_t$ for the frame $t$. Finally, the speech token embeddings $\mathbf{E}_t$ are fed into the LM-style transformer models for further processing.

We also implemented other popular speech tokenizers including HuBERT-KM (Lakhotia et al., 2021) and SpeechTokenizer (Zhang et al., 2023b) for comparison. The main difference among these speech tokenizers is the codebook size and codes dimension, shown in Table 2. For both HuBERT-KM and SpeechTokenizer the speech tokens are mapped via a learnable linear layer from their dimension to the text embedding dimension $D$ before feeding into the LM-style transformer model. In Table2, we also compare dMel with the baselines in terms of vocabulary size, bit-rate, and

Table 2: Comparison between different speech tokenizers: dMel (ours), HuBERT-KM and Speech-Tokenizer. For dMel we use $N = 80$ log mel-filterbanks (50ms window, 25ms hop distance), and $2^K = 16$ values of the codebook $\mathbf{C}$. For HuBERT-KM, 200 is chosen accoring to Maiti et al. (2024).

|  | dMel | HuBERT-KM | SpeechTokenizer |
|---|---|---|---|
| Codebook Size | 16 | 200 | 1024 |
| Code Dimension | 80 | 1 | 8 |
| Vocab Size | 16 * 1 | 200 * 1 | 1024 * 8 |
| Frame-rate | 40Hz | 50Hz | 50Hz |
| Bit-rate | 12.8kps | 0.4kps | 4kps |
| Training-free? | ✓ | ✗ | ✗ |

frame-rate. First, `dMel` has a much smaller vocabulary, as it is discretized mel-filterbanks energies, allowing all 80 channels to share the same vocabulary since they represent similar energy values. In contrast, neural compression encoders like SpeechTokenizer require separate embeddings for different channels. Also, `dMel` operates at a lower frame-rate while maintaining a higher bit-rate. The reduced frame-rate leads to shorter sequence lengths during both training and inference, which is particularly advantageous when using large models. While a higher bit-rate typically increases model complexity for compression-based tokenizers, this is not the case for `dMel` due to two key factors: i) dMel is encoder-free, without any compression encoder; ii) the complexity of the model introduced in Section 2.2 depends only on the vocabulary size and sequence length (frame-rate), not on the code dimensions. In compression-based methods, increasing the bit-rate requires either larger codebooks or additional residual dimensions, leading to increased tokenizer complexity. Moreover, these methods require more complex downstream models to handle the expanded representations. Given recent studies (Défossez et al., 2024; Mousavi et al., 2024) demonstrated that bit-rate does not strongly correlate with downstream model performance, we focus our comparative analysis on frame-rate rather than bit-rate when evaluating different speech tokens for downstream tasks. This approach challenges the conventional assumption that higher bit-rates necessarily yield better results.

**Speaker Representation**    To properly model multi-speaker data, we also include speaker embeddings as input to the transformer decoder. The speaker embeddings are extracted from an independent dvector (Variani et al., 2014) model[3]. We use a learnable linear layer to map the speaker embeddings to the same dimension as the speech and text token embeddings $D$. The speaker representation is optional for ASR task, but required for TTS task. Hence, during the training, it is applied for text-to-speech and ignored for speech-to-text.

**Transformer Decoder**    The transformer decoder is trained end-to-end on a combined dataset of speech and text pairs. For TTS training, the input sequence is constructed by concatenating the speaker embedding (extracted from a random audio for the same speaker of the current sample), text tokens, and speech tokens. For ASR training, the input sequence is constructed by concatenating the speech tokens and text tokens. Both tasks are trained with causal masking, where the model is trained to predict the next token based on the previous tokens. The loss is calculated using the cross-entropy loss between the predicted tokens and the ground-truth tokens. Loss calculation is skipped on the speech tokens for ASR task and on the text tokens for TTS task. ***Note, that all frequency channels at time frame $t$ for `dMel` tokenizer are predicted independently and in parallel.*** To capture the relative distances between tokens in the input sequence, we apply multiplicative relative positional embedding RoPE (Su et al., 2024). This allows the model to learn the positional relationships between speech tokens, text tokens, and speaker embeddings, enhancing its ability to generate coherent output sequences. For positional embeddings we do not distinguish between text, speech and speaker tokens and thus having global positions notation across all of them, see Figure 2.

**Robust Training**    Compared to LMs, audio frames are highly redundant with strong local correlations. This makes longform generation difficult for models due to exposure bias (Bengio et al., 2015). To mitigate exposure bias during training, we apply span-masking (Raffel et al., 2020) to the speech token context, masking out multiple random spans of speech frames. The model is trained to predict the next token based on the masked context. This context-masking strategy helps the model learn to generate accurate speech tokens in the presence of missing information, improving its robustness and generalization. It forces the model to attend to the text rather than copying previously inferred speech tokens due to learnt correlations. We also find that span-masking text tokens improves the ASR task.

## 3 EXPERIMENTS

In this section, we begin by evaluating different speech tokenizers through a common practice in the literature: tokenizing speech into discrete units and then reconstructing the speech to assess the quality of the reconstruction. This approach helps gauge the effectiveness of various tokenization techniques. Following this, we present both TTS and ASR results using an LM-style (decoder-only) model with different speech tokens. While most related work focuses solely on speech synthesis, our study encompasses both speech generation and recognition, providing a more comprehensive evaluation of the tokenization methods. We evaluate the performance of our model mainly on the LibriSpeech dataset and compare it with state-of-the-art speech tokenizers, ASR and TTS models.

---

[3]We use a pretrained model "Speaker Encoder" from the YourTTS (Casanova et al., 2022) repository `https://github.com/Edresson/YourTTS`.

Table 3: Speech reconstruction results on 300 random samples from LibriSpeech *test-clean* set.

| Tokenizer | Speech2Unit (M params) | Unit2Speech (M params) | Frame Rate | WER↓ | MOS-LQO↑ | MOS↑ (95% CI) |
|---|---|---|---|---|---|---|
| GroundTruth | - | - | - | 2.02 | - | 3.91±0.12 |
| HuBERT-KM | 95 | 111 | 50Hz | 8.71 | 2.06 | 2.74±0.14 |
| EnCodec | 7 | 7 | 75Hz | 2.03 | 4.03 | 3.69±0.13 |
| SpeechTokenizer | 65 | 34 | 50Hz | 2.41 | 4.19 | 3.77±0.13 |
| Mel-HifiGAN | n/a | 12 | 80Hz | 2.08 | 4.52 | 3.80±0.12 |
| dMel-HifiGAN | n/a | 12 | 80Hz | 2.11 | 4.47 | 3.68±0.13 |
| Mel-PWG | n/a | 1 | 80Hz | 2.13 | 4.40 | 3.27±0.14 |
| dMel-PWG | n/a | 1 | 80Hz | 2.23 | 4.37 | 3.23±0.14 |
| Mel-PWG | n/a | 1 | 40Hz | 2.36 | 4.34 | 2.99±0.15 |
| dMel-PWG | n/a | 1 | 40Hz | 2.51 | 4.29 | 2.97±0.15 |

## 3.1 TRAINING DATA

We use several open-sourced datasets with paired speech and text transcription to conduct experiments: i) LibriSpeech (Panayotov et al., 2015) dataset (CC BY 4.0) consists of English speech recordings (960h, 16kHz) from various speakers (∼2k) and conditions; ii) LibriTTS (Zen et al., 2019) (CC BY 4.0) dataset (500h) derived from LibriSpeech improves on it with the proper sentence split, text normalization and keeping samples 24kHz; iii) VCTK (Yamagishi et al., 2019) contains 44h of English speech (108 speakers); iv) LJSpeech (Ito & Johnson, 2017) (public domain in US) is a single speaker English audio recordings of 16kHz with read speech from LibriVox[4]. While LibriSpeech is used to train ASR and TTS models, LibriTTS, VCTK and LJSpeech are only used to train the TTS.

## 3.2 TRAINING CONFIGURATION

We train the LM-style transformers in three different sizes: Small, Base, and Large (see Appendix Table 11). Unless stated otherwise, the Base model is used in all experiments if not stated otherwise. All models use pre-LayerNorm with dropout set to 0.1 for residual, attention and embedding layers and 0.3 for positional embedding. dMel uses 16 discrete bins for each channel while text is tokenized with a character vocabulary; the speaker embedding dvector has 512 dimensions (see Appendx E for details). In all experiments, training data are sampled to 16kHz.

## 3.3 MAIN RESULTS

### 3.3.1 SPEECH RECONSTRUCTION

Following Zhang et al. (2023b), we randomly sample 300 speech utterances and their ground truth transcriptions from the LibriSpeech *test-clean* dataset. We use the speech2unit and unit2speech modules to convert the speech signal to speech tokens and then reconstruct the speech signal from the speech tokens. We compute the WER between the ASR outputs from HuBERT-Large (Hsu et al., 2021)[5] on the audio samples and their ground truth transcripts. We also report MOS-LQO (Mean Opinion Score – Listening Quality Objective) score to measure the reconstruction quality using ViSQOL (Hines et al., 2012). Finally, we use human evaluation to measure the naturalness of the reconstructed speech using a MOS score with 95% confidence interval. We instruct the human evaluators to rate the naturalness of the reconstructed speech on a scale of 1 to 5, where 1 is the worst and 5 is the best. The results are shown in Table 3.

From Table 3, we can see that semantic tokenization (HuBERT-KM) is not good for speech reconstruction. Meanwhile, acoustic tokenizers that are optimized to reconstruct the signal directly (EnCodec and SpeechTokenizer) do well.

We apply different vocoders to reconstruct the speech signal from log mel-filterbanks, and find that the WER of the reconstructed speech signal is comparable to the acoustic tokenization methods with a fraction of the parameters. Also, log mel-filterbanks achieve a better MOS-LQO score, which indicates that the reconstructed audio is more similar to the original audio. By comparing Mel and dMel, we can see that discretization has little impact on WER and MOS-LQO scores. We also find

---

[4]https://librivox.org/pages/public-domain/.

[5]We use checkpoint https://huggingface.co/facebook/hubert-large-ls960-ft.

Table 4: Text-to-speech results for different tokenizers. `RichTTS` is trained on LibriSpeech 960h. WER (%) and CER (%) are evaluated with WhisperX ASR ("base.en") and reported on *test-clean*.

| Model | WER↓ (%) | CER↓ (%) | Params |
|---|---|---|---|
| VOXTLM (HuBERT+KM)†, Maiti et al. (2024) | - | 3.5 | 350M |
| USLM (SpeechTokenizer)†, AR+NAR, Zhang et al. (2023b) | 6.5 | - | 356M |
| RichTTS (HuBERT+KM) | 9.5 | 4.3 | 258M |
| RichTTS (SpeechTokenizer), AR | 11.4 | 5.9 | 258M |
| RichTTS (dMel) | **4.3** | **1.8** | 258M |

Table 5: WER (%) (evaluated with WhisperX ASR "base.en") and MOS of different TTS models' generations using transcriptions from each evaluation set that correponds to data used for training.

| | WER↓ (%) | | | MOS↑ (95% CI) |
|---|---|---|---|---|
| Model | LJSpeech | LibriTTS | VCTK | VCTK |
| GroundTruth | 2.6 | 3.8 | 3.4 | 4.18±0.10 |
| Tacotron2, Shen et al. (2018) | 4.4 | 7.3 | 4.2 | 2.91±0.15 |
| FastSpeech2, Ren et al. (2020) | 6.1 | 10.2 | 3.8 | 3.03±0.14 |
| VITS, Casanova et al. (2022) | 6.4 | 8.3 | 11.1 | **3.56±0.12** |
| RichTTS (dMel) | **4.0** | **4.5** | **2.2** | 3.34±0.14 |

that the exact vocoder matters much less than the frame rate of tokenization: the WER goes from 2.08 to 2.13 when switching from HifiGAN to ParallelWaveGAN, but it falls from 2.13 to 2.36 when the frame rate is changed from 80Hz to 40Hz. However, even a 1M parameter vocoder operating at a 40Hz frame rate is comparable to the much larger SpeechTokenizer on WER and MOS-LQO metrics.

*Considering the efficiency and performance, we choose the `dMel` speech tokenizer in 40Hz with ParallelWaveGAN vocoder for the following experiments.*

### 3.3.2 LM-STYLE TEXT-TO-SPEECH

Here we compare the accuracy and naturalness of speech synthesized by LM-style text-to-speech (TTS) models trained on different tokenization methods. For TTS evaluation, we utilize WhisperX (Bain et al., 2023) ("base.en" from Radford et al. (2023)) to transcribe our generated speech into text and calculate the WER and the character error rate (CER). We report both WER and CER to facilitate comparisons to prior works which have reported only one or the other.

We trained the TTS model using the same architecture but with three different tokenization methods: HuBERT+KM (with 200 clusters), SpeechTokenizer, and `dMel`. Additionally, we present the results from VOXTLM (Maiti et al., 2024) and USLM (Zhang et al., 2023b) for comparison. VOXTLM is a larger model trained on more data that is initialized from a pretrained LLM (OPT) using HuBERT-KM as the speech tokenizer. USLM comprises an autoregressive (AR) model and a non-autoregressive (NAR) model, both trained with the SpeechTokenizer.

As shown in Table 4 for training on LibriSpeech dataset, our LM-style model with `dMel` tokenization achieves a WER of 4.3 and a CER of 1.8, significantly outperforming the baseline methods. This indicates that our model can generate more accurate speech with less hallucination and distortion. Furthermore, we observed that the AR model trained on SpeechTokenizer tokens exhibits a much higher WER compared to the idiosyncratic coarse to fine models (labeled AR+NAR) developed for these residual tokenizers – indicating that `dMel` lies on a simpler data manifold.

Given the success of our LM-style `dMel` TTS model, dubbed `RichTTS`, we further evaluate it on various datasets, including LJSpeech, VCTK, and LibriTTS, and compare it with popular open-sourced TTS models, including Tacotron2 (Shen et al., 2018), FastSpeech2 (Ren et al., 2020), and VITS (Kim et al., 2021). We conduct human evaluation to measure the naturalness of 50 randomly sampled synthesized speech from VCTK test set. `RichTTS` achieves competitive performance on the TTS task in terms of both MOS and WER demonstrating its effectiveness in generating high-quality synthesized speech, see Table 5. Interestingly, we find that VITS performs poorly on the VCTK WER. We suspect this is because VITS tends to make more mistakes at the beginning of each sequence, and since VCTK comprises short sequences, even one or two word errors can lead to a high WER.

Furthermore, we observed that our model with `dMel` tokenization can generate long audio sequences with high quality. Here, we evaluate the performance of our model on different lengths of text

Table 6: results for TTS models trained on LibriSpeech 960h and evaluated on LJSpeech test set.

| Sequence Length | Tacotron2 | FastSpeech2 | VITS | RichTTS |
|---|---|---|---|---|
| Total WER↓ (%) | 4.4 | 6.1 | 6.4 | **4.0** |
| 10-20 words | 5.5 | **3.1** | 7.4 | 3.5 |
| 20+ words | 3.3 | 9.1 | 5.3 | **3.0** |

Table 7: Speech recognition results for different tokenizers measured with WER (%). All models are trained on LibriSpeech 960h.

| Model | dev-clean↓ | dev-other↓ | test-clean↓ | test-other↓ | Params |
|---|---|---|---|---|---|
| RichASR (SpeechTokenizer) | 6.5±0.3 | 16.9±0.7 | 6.9±0.4 | 17.5±0.5 | 258M |
| RichASR (HuBERT+KM) | 5.3±0.1 | 13.7±0.2 | 5.8±0.1 | 13.8±0.1 | 258M |
| RichASR (dMel) | **3.8**±0.1 | **10.3**±0.1 | **4.2**±0.2 | **10.4**±0.1 | 258M |

Table 8: Comparison of WER (%) for best RichASR trained with dMel tokenization and prior work with LM-style ASR models and HuBERT+KM with subword modeling on top as tokenization.

| Model | Data (h) | dev-clean↓ | dev-other↓ | test-clean↓ | test-other↓ | Params |
|---|---|---|---|---|---|---|
| VOXTLM | 280k | - | - | 6.5 | 17.6 | 350M |
| VOXTLM | 280k | - | - | 4.6 | 12.1 | 1.3B |
| Chen et al. (2024) | 960 | 3.6 | **7.8** | 3.8 | **8.3** | 355M |
| RichASR (dMel) | 960 | **3.1** | 8.4 | **3.4** | 8.6 | 355M |

sequences using the LJSpeech test set. Table 6 shows the WER results for our model on text sequences with 10-20 words and more than 20 words. We ignore text sequences with fewer than 10 words, as they are too short and not robust for WER evaluation. From Table 6, we observe that our model achieves competitive performance across different text lengths, demonstrating its robustness and generalization ability in generating synthesized speech for varying text inputs lengths. Additionally, we find that the non-autoregressive (NAR) model FastSpeech2 achieves the lowest WER on shorter sequences but the highest WER on longer sequences. This suggests that NAR models may not be well-suited for generating long audio sequences.

### 3.3.3 LM-STYLE SPEECH-TO-TEXT

Training an LM-style speech-to-text (ASR) model can test if the speech tokens can preserve the semantic information in the speech signal and support the speech content-based task. Table 7 shows results of our model dubbed RichASR, trained with different tokenizations including dMel for the ASR task. Our LM-style model with dMel speech tokenization achieves 4.2% WER on the *test-clean* and 10.4% WER on the *test-other* sets outperforming both HuBERT-KM and SpeechTokenizer. We also observe that our model with HuBERT-KM (Lakhotia et al., 2021) outperforms the SpeechTokenizer (Zhang et al., 2023b) for ASR, which is reasonable as semantic tokens are more suitable for the ASR task.

In Table 8, we further compare RichASR with dMel speech tokenizer trained with GPT-2-meduim architecture (Radford et al., 2019) on LibriSpeech 960h with prior work: VOXTLM (Maiti et al., 2024) that uses larger model trained with more data and initialized from a pretrained LLM (OPT Zhang et al. (2022)), and HuBERT-KM with additional subword modeling on top as the speech tokenizer; Chen et al. (2024) that also uses GPT-2 architecture trained on LibriSpeech 960h and HuBERT-KM with additional subword modeling on top as the speech tokenizer[6]. RichASR with dMel outperforms VOXTLM; it also outperforms Chen et al. (2024) on clean sets and a bit behind it on other sets.

The ASR results clearly demonstrate the benefit of using our dMel speech tokenizer for the content-related tasks in speech, as it better preserves the semantic information in the speech signal. Further details and ablations can be found in Appendix E and F.

### 3.3.4 ABLATIONS

We first investigate the impact of the codebook sizes, shown in Table 9. The 16-bin configuration used in the paper demonstrates the best overall performance across tasks. While the 32-bin setup

---

[6]We use official codebase to train this model w/o text pretraining as Chen et al. (2024) report results only with text pretraining.

Table 9: ASR and TTS results (WER, %) with `dMel` speech tokenizer and different number of bins (codebook size) for discretization in `dMel`. All models are trained on LibriSpeech 960h.

| N-Bins | ASR test-clean↓ | ASR test-other↓ | TTS WER↓ (%) |
|---|---|---|---|
| 8 | 6.6 | 16.5 | 7.3 |
| 16 | 4.4 | 10.7 | 4.8 |
| 32 | 4.7 | 10.2 | 5.7 |

slightly outperforms on the ASR *test-other* set, it shows degraded performance in TTS. This trade-off likely stems from the increased speech vocabulary size, which may pose challenges for accurate prediction. The results may get better with increased data and model size. And 8-bin configuration looses too much information with discretization.

We then ablate the ASR results to understand why ASR LM-style model is behind the state-of-the-art on LibriSpeech. We take two existing transformer ASR baselines, Seq2Seq and CTC, that use 80 log mel-filterbanks and characters as targets. We then modify these baselines by using `dMel` instead (the discretization, embedding layer and linear layer) while keeping all other hyper-parameters the same (we adjust only the SpecAugment time masking max width accordingly to keep total masking in *ms* the same). Our results (Appendix Table 13) suggest: i) `dMel` brings only small degradation compared to Mel; ii) additional discrepancy is coming from different hop distance in featurization; iii) *the main and significant performance degradation is coming from switching to LM-style model.* The latter is in line with Maiti et al. (2024) and Chen et al. (2024), though was not discussed in detail by any prior work. We hypothesise this gap is due to observed overfitting of the LM-style models.

### 3.3.5 UNLOCKING JOINT SPEECH-TEXT MODELING

Our model design allows us to train a single model for both ASR and TTS tasks leading to a simpler setup. We train a single model with the same architecture and tokenization as `RichTTS`, by constructing the training data with <text, speech> and <speech, text> pairs for ASR and TTS tasks, respectively. By mixing these two types of data, we can train a single model for both tasks.

Table 10 shows that the joint model is worse on both tasks, but ASR is affected more than TTS. Comparing our results to VOXTLM, which initializes its model from pretrained LLM (OPT) and finetunes it with multiple tasks and datasets, we speculate that our joint model needs text-only training to learn a good LM for better ASR performance. Our model structure trivially allows for this text-only training, but we leave those experiments for future work (for further discussion see Appendix F.3).

## 4 RELATED WORK

**Speech Tokenization** Recent advancements in speech tokenization have primarily focused on two approaches: semantic tokens and acoustic tokens. This section examines these methods, their combinations, and their limitations, highlighting the need for more efficient and generalizable solutions. Semantic tokens, extracted from self-supervised pretrained speech models, have shown promise in capturing high-level content information. Methods like wav2vec (Baevski et al., 2020) and HuBERT (Hsu et al., 2021) employ $k$-means clustering on speech representations to generate these tokens. While effective in capturing semantic content, these approaches often struggle with preserving fine-grained acoustic details crucial for high-quality speech synthesis. In contrast, acoustic tokens, derived from pretrained audio compression models, excel at preserving low-level acoustic information. Techniques such as SoundStream (Zeghidour et al., 2021) and EnCodec (Défossez et al., 2022) utilize residual vector quantization (RVQ) with reconstruction objectives. These methods achieve high-quality audio compression but may not capture higher-level semantic structures effectively.

Recognizing the complementary nature of semantic and acoustic tokens, recent works have attempted to combine these approaches. AudioLM (Borsos et al., 2023) introduced a three-stage model: semantic modeling, coarse acoustic modeling, and fine acoustic modeling. While comprehensive, this approach introduces complexity and computational overhead. AudioPalm (Rubenstein et al., 2023) further demonstrated the critical importance of large-scale training data and model parameters for effective multi-stage modeling, highlighting potential generalization issues in low-resource scenarios. An alternative hybrid approach, proposed by Zhang et al. (2023b), attempts to distill semantic

Table 10: Results of ASR and TTS jointly trained model on LibriSpeech 960h.

| | ASR, WER↓ (%) | | TTS | |
|---|---|---|---|---|
| | test-clean | test-other | WER↓ (%) | CER↓ (%) |
| VOXTLM+OPT, 350M | 3.5 | 8.7 | - | 3.5 |
| `RichASR-RichTTS`, single models, 258M | 4.2 | 10.4 | 4.3 | 1.8 |
| `RichASR-RichTTS`, joint model, 258M | 7.6 | 20.0 | 4.4 | 1.9 |

information into acoustic tokens during RVQ model training. However, this method still requires additional pretraining and does not fully achieve a single-stage model architecture.

Despite these advancements, several challenges persist in the field of speech tokenization: i) balancing semantic and acoustic information in a unified representation; ii) reducing model complexity and computational requirements; iii) improving generalization to low-resource / out-of-domain data e.g. with mixed speech from multiple speakers, or multiple languages, or changing characteristics of recording equipment/sampling rate etc.; iv) developing truly single-stage tokenizers. Our proposed method, `dMel`, addresses these challenges by offering a training-free speech tokenization approach. By directly discretizing log mel-filterbanks into bins, it inherently preserves both semantic and acoustic information in a unified representation, while significantly reducing computational complexity. Concurrently, Langman et al. (2024) proposed a different mel-filterbanks based speech tokens: spectral codecs, where disjointed mel-bands are encoded separately and then quantized using an FSQ (Mentzer et al., 2023). Although spectral codecs and `dMel` are both discretizing mel-filterbanks, `dMel` is encoder-free and tested with autoregressive generation tasks, while spectral codecs need encoders and is tested with non-autogressive generation task. Also, our scalar quantization method shares some similarities with FSQ, but FSQ is in a learned latent code space and necessitates an additional bound operation to limit the range of the latent codes.

**Speech-Text Modeling**   Modeling speech and text jointly is a challenging task, as speech signals are continuous and while text is discrete. Existing works have explored various approaches to address this challenge, including usage of separate encoders for different modalities (Ao et al., 2021; Bapna et al., 2021). Bai et al. (2022) proposed an encoder-only model A3T for speech-text modeling, by introducing alignment embedding to encourage cross-modal transfer between text and speech. Although A3T achieved good performance on speech synthesis and editing tasks, it cannot generate text and cannot generalize to longform generation because of its encoder-only architecture and mask-reconstruction training strategy. VioLA (Wang et al., 2023b) also targets a unified speech-text model which can generate speech and text with a single model, but it is specifically designed for the Encodec (Défossez et al., 2022) style feature, and compelled to model speech tokens in a multi-stage hierarchical manner. Maiti et al. (2024) proposed a LM-style model VOXTLM, to model speech and text jointly. However, VOXTLM is only models the HuBERT semantic tokens, and relies on an external generation model to transform semantic tokens into waveform, but the speaker and acoustic information are lost. In comparison, the model architecture in this paper is a simple, single stage LM-style transformer model, and can handle both the speech generation and text generation tasks.

## 5 CONCLUSION

In this work, we proposed `dMel`, a novel train-free speech tokenization method that discretizes log mel-filterbank energies directly into bins. By operating on the authentic log mel-filterbank representation, `dMel` inherently preserves both semantic and acoustic information in a unified tokenized representation. Our key contribution is the evaluation of `dMel` within a unified LM-style transformer architecture for speech recognition (ASR) and speech synthesis (TTS) tasks. Our `dMel`-based ASR model, `RichASR`, achieved the lowest word error rate among tokenization methods, robustly preserving semantic content. For TTS, `dMel`'s generation yielded the lowest WER, accurately reconstructing speech waveforms. Our `dMel`-based TTS model, `RichTTS`, achieved competitive naturalness, lowest error rates, and long audio generation capabilities.

`dMel`'s simplicity circumvents separate tokenizers or multi-stage modeling, reducing computational overhead and dependence on pretrained models. By unifying semantic and acoustic modeling, `dMel` enables efficient speech-text modeling frameworks. While initial joint TTS-ASR training showed promise, further work is needed. Our primary contribution demonstrates `dMel`'s effectiveness for high-performing separate TTS and ASR models within a unified LM-style architecture.

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

## A    ETHICS STATEMENT

The development and deployment of speech technologies carry important ethical considerations. While our proposed `dMel` method aims to advance the state-of-the-art in speech-text modeling, it is crucial to highlight potential ethical risks and raise the awareness so that new methods may be developed to mitigate these risks.

Our first main concern is the potential dual-use of speech synthesis technologies for nefarious purposes such as impersonation, misleading audio-visual content generation, or voice spoofing attacks. Proactive measures, including watermarking techniques and robust speaker verification methods, should be explored to counter such risks. The former attempts to build markers into the generated speech that make it easy to detect, while the latter focusses on distinguishing synthetic from real data. Prior work (Le et al., 2023) has shown that neural networks can be trained to distinguish speech synthesized from their model from real speech, probably because of artifacts from the use of mel spectral vocoders. While we did not train a network to do so in our work yet (we will create one before code release), the vocoders we use are similar to their work – going from mel spectrogram to raw waveforms. Our model also does not use prosody, phoneme duration and other predictions that more sophisticated TTS systems use to allow the model to perform very well on imitating speaker styles in zero-shot settings. However our model can probably mimic the styles of training speakers very well. It is our hope that releasing our methods will facilitate more research on fake speech verification and watermarking techniques – even if current classifiers are able to perform this detection, the quality of the generative models is improving. It is also our hope that future works will attempt to perform more credit assignment – by providing metrics that show which real data samples a synthetic speech example copies its style and substance from.

Another concern is the perpetuation of societal biases encoded within training data. Speech datasets may exhibit biases along dimensions such as gender, race, age, or socioeconomic status, which could be propagated or amplified by trained models. Rigorous debiasing techniques and careful curation of representative training data are essential to mitigate these risks. On the mitigating side of this equation, we also hope that with better, more controllable TTS systems, ASR systems can improve because more data can be generated for underrepresented segments of the distribution from the TTS models.

Furthermore, the development and deployment of speech technologies should prioritize accessibility and inclusivity. Models should be evaluated for performance across diverse demographics, accents, and language varieties to ensure equitable access and quality of service.

Finally, it is important to foster transparency and accountability in the research and development process. Clear documentation of model capabilities, limitations, and potential failure modes should be provided to enable informed decision-making and responsible usage.

Addressing these ethical considerations requires a multistakeholder approach involving researchers, developers, policymakers, and end-users. By prioritizing ethical principles such as fairness, privacy, and accountability, we can work towards realizing the benefits of speech technologies while mitigating potential risks and adverse societal impacts.

## B    LIMITATIONS

Because TTS work is tremendously fragmented and clear protocols are not often available for training and evaluation, we reimplemented other tokenizers within our code base using publicly available, official implementations where available: e.g. we used Hubert-KM and speech tokenizer features extraction from the public codebases and pluged them into our LM-style model training. While we made the best effort to tune the tokenization methods and the models, there is always a possibility we missed some details. However, our results seem to tell a consistent story when viewed from multiple angles, and when viewed on multiple datasets. We also did not train on larger model sizes (>1B parameters), larger datasets (>1k hours), or using pretrained models.

The real challenge for modern multimodal LLMs is complex semantic understanding tasks. While our current experiments focus on text-to-speech and speech-to-text tasks, these encompass critical aspects of speech processing. `dMel`'s effective performance within a decoder-only architecture for both tasks suggests potential for broader applications. We recognize the importance of more sophisticated

speech understanding tasks and view our work as a foundation for future research leaving other tasks out of scope of the paper. Scaling up pretraining and exploring complex semantic understanding tasks could further validate our approach's versatility across a wider range of multimodal language processing challenges.

We acknowledge that our current scope targets only speech on purpose, as indicated in our title. While `dMel` may potentially support non-speech tasks, ***our current exploration and verification focus solely on speech, not general audio.*** Regarding the "speaker variations" – mel-spectrogram is used for speaker recognition widely, thus it preserves necessary speaker information on which we thus rely in `dMel` too.

## C    DATA, CODE, REPRODUCIBILITY

We made the best effort to use publicly available data and official implementations of prior works where it is possible. All data we used are under permissive license for research. We provided as much as detail as is possible without code such as details on our model training and hyperparameters throughout the paper and in the Appendix. We plan to open-source our code upon paper acceptance.

We do not plan to open-source any pre-trained models for sake of privacy, safety and misuse.

## D    SUBJECTIVE EVALUATION FOR TTS

We use crowd-sourcing to collect subjective ratings to compare the naturalness of the reconstructed speech from the different tokenizers. We evaluate the quality of the same (randomly sampled) 50 utterances for each model by collecting around seven ratings per sample. Overall, we collect 3500 ratings from 65 raters. The raters were English-speaking and were paid at least the minimum wage.

We present the raters with a generated speech sample and instruct them to rate how natural it sounds on a five-point Likert scale, where 1 corresponds to very unnatural and 5 corresponds to very natural. Figure 3 shows a screenshot of our subjective test as seen by the rater.

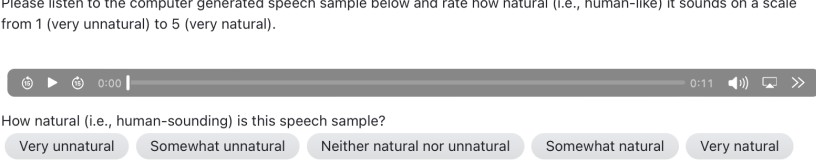

Figure 3: A screenshot of the assessment task, as the crowd-sourced rater sees it.

We noticed human annotators have bias over audio volume so we do volume normalization on top of all reconstructed or generated audio before giving them to human annotators.

We report Mean Opinion Score (MOS) results throughout the paper with confidence intervals calculated using bootstrap resampling with 1000 iterations, providing a reliable estimate of the variability MOS results.

## E    TRAINING DETAILS

### E.1    BASELINES

For reproducibility, we provide the HuggingFace model cards used in our experiments in Table 5:

- Tacotron2 (Shen et al., 2018), `https://huggingface.co/espnet/espnet/kan-bayashi_vctk_tts_train_xvector_tacotron2_raw_phn_tacotron_g2p_en_no_space_train.loss.ave`
- FastSpeech2 (Ren et al., 2020), `https://huggingface.co/espnet/kan-bayashi_vctk_gst_fastspeech2`

- VITS (Casanova et al., 2022), `https://huggingface.co/espnet/kan-bayashi_vctk_multi_spk_vits`

## E.2 RICHASR AND RICHTTS

For our LM-style model we stack together speaker embedding, speech tokens and text tokens. Both speech and text tokens have prepended begin of sentence token (<bos>) and appended end of sentence token (<eos>).

We train all models using the Adam optimizer with a learning rate of 1e-3, learning rate warmup of 4k steps for ASR and 5k for TTS, cosine learning rate schedule and gradient clipping of 1.0 for TTS and 0.1 for ASR and joint models. We use dynamic batching to optimize the data packing with total batch size of 1.4h/1.4h/0.7h for ASR training and 1h/2h/2h for TTS training for Small/Base/Large models. We train TTS models for 100k steps and ASR models 80k steps with mixed precision training and BF16 on A100 and H100 GPUs with 80GB. Both ASR models and TTS models are trained with 8GPUs for less than a day and for 2-4 days for ASR and TTS respectively.

Table 11: LM-style transformer model configurations for ASR, TTS and joint models training.

|  | Small | Base | Large |
|---|---|---|---|
| # of layers | 18 | 36 | 48 |
| # of attention heads | 2 | 4 | 8 |
| # of hidden units ($D$) | 512 | 768 | 1536 |
| # of parameters | 59M | 258M | 1.3B |

## E.3 LM-STYLE SPEECH-TO-TEXT

For ASR training as an augmentation we apply SpecAugment (Park et al., 2019) with 2 frequency masks with max width 30 and 10 time masks with max width 50 and ratio 0.1. With ablations we found that SpecAugment masking with average value instead of zero is slightly better. Without applying SpecAugment performance of ASR is 7.3% WER on *dev-clean* and 20.3% WER on *dev-other*, which is further can be improved with usage of frequency masking only to 6.4% WER on *dev-clean* and 16.6% WER on *dev-other*. Usage of both frequency masking and time masking results in the best performance of Table 7.

We found that span masking is key part of model training to enforce self-attention to attend to speech part as well as to reduce exposure bias. The masking strategy is similar to the one used for TTS training: for every training step with probability $p$ the sample in the minibatch is masked with the mean span of 3 tokens with masking ration of 0.5. We found that the mean span of 1 token or 5 tokens gives the same results; while the mask probability $p$ is the most important hyper-parameter. The optimimal value for ASR is found to be 0.8, which is used in all final models.

As we found one best model configuration for the Base model with `dMel` we then change only i) model size ii) speech tokenization iii) training data (here we increase model dropout to 0.3 for training on *train-clean-360* and to 0.5 for training on *train-clean-100* as otherwise models drastically overfit); the rest of hyper-parameters stay the same.

## F ABLATIONS

### F.1 LM-STYLE TEXT-TO-SPEECH

Scaling results for `RichTTS` are shown in Table 12.

### F.2 LM-STYLE SPEECH-TO-TEXT

ASR ablations for different model sizes, data sizes, and tokenizers are shown in Table 14.

We noticd the results in Chen et al. (2024) seems to be the SOTA for LM-style ASR model to the best of our knowledge. However, as many ablations are missed in Chen et al. (2024), we took their

Table 12: Text-to-speech results for different model sizes with `dMel`. All models are trained on LibriSpeech 960h dataset. Evaluation is done via speech generation on the full *test-clean* transcriptions and speakers, and then evaluated WER with WhisperX base.en.

|  | WER↓ (%) |
|---|---|
| `RichTTS` (`dMel`), Small | 8.1 |
| `RichTTS` (`dMel`), Base | 4.3 |
| `RichTTS` (`dMel`), Large | 5.4 |

Table 13: WER (%) comparison for CTC, Seq2Seq, and LM-style ASR models (∼260M) trained on LibriSpeech 960h with `dMel` and Mel features. We compute 80 log mel-filterbanks with 25ms (50ms) window and 10ms (25ms) hop distance, denoted as '10ms' ('25ms').

| Model | Features | dev-clean↓ | dev-other↓ |
|---|---|---|---|
| Gulati et al. (2020) (RNN-T – Conformer) | Mel-10ms | 1.9 | 4.1 |
| Kim et al. (2022) (CTC – Squeezeformer) | Mel-10ms | 2.3 | 5.8 |
| | Mel-10ms | 2.4 | 5.4 |
| Seq2Seq (Dong et al., 2018) | `dMel`-10ms | 2.5 | 5.9 |
| | Mel-25ms | 2.8 | 6.5 |
| | `dMel`-25ms | 2.7 | 6.2 |
| | Mel-10ms | 2.1 | 5.4 |
| CTC (Graves et al., 2006) | `dMel`-10ms | 2.1 | 5.6 |
| | Mel-25ms | 2.1 | 5.4 |
| | `dMel`-25ms | 2.3 | 6.1 |
| LM-style | `dMel`-25ms | 3.4 | 9.5 |

open-sourced code and run ablations ourselves to have proper comparison with it. The final results, including ablation with `dMel` are shown in Table 15:

- We successfully reproduced Chen et al. (2024) results (row 1 and 2).

- Without pretraining (rows 3, 4, 5):

  `dMel` outperforms HuBERT-KM on both clean and other datasets; `dMel` surpasses BPE on top of HuBERT-KM on clean data, while BPE on HuBERT-KM performs better on other.

- Without pretraining and without speed perturbation (rows 6, 7, 8):

  BPE on HuBERT-KM performance decreases significantly after diabling speed perturbation (compare rows 3 and 6), raising questions about its generalizability to other domains, given that BPE tokens are trained on speed-perturbed LibriSpeech data.

  Our `dMel` (row 8) achieves substantially better results than both HuBERT-KM and BPE on HuBERT-KM (rows 7 and 6), demonstrating robust performance even without speed augmentation.

Note that in `dMel`, we use SpecAugment (masking across time and channels) and Chen et al. (2024) also use SpecAugment. According to their code, the time masking is 30%, while channel masking is impossible as there is only 1 channel).

We believe these results demonstrate the effectiveness, simplicity in use, and robustness of our `dMel` tokenization method, particularly in scenarios where extensive pretraining or domain-specific augmentations may not be feasible.

Note that Chen et al. (2024) did not show applicability of BPE on HuBERT-KM or HuBERT-KM to TTS task, while in VOXTLM (also uses BPE on HuBERT-KM) it is shown that this tokenization is not suited for TTS (the performance is poor). `dMel` in contrary is shown to perform well on TTS task too in addition to ASR.

Table 14: Our ASR models trained on different subsets (*train-clean-100* LS-100, *train-clean-360* LS-360, full LibriSpeech LS-960) of LibriSpeech, with different model sizes and different speech tokenizations (greedy decoding is reported). Results are shown across 2 runs with mean WER and standard deviation.

| Tokenization | Model Size | Data | dev-clean↓ | dev-other↓ | test-clean↓ | test-other↓ |
|---|---|---|---|---|---|---|
| dMel | Base | LS-100 | 18.1±1.0 | 39.4±1.2 | 19.0 ±1.0 | 41.3±1.1 |
| dMel | | LS-360 | 6.4±0.4 | 20.1±1.1 | 6.9 ±0.6 | 20.5±0.9 |
| SpeechTokenizer (Zhang et al., 2023b) | Small | LS-960 | 6.2±0.2 | 16.8±0.3 | 6.5±0.2 | 17.4±0.3 |
| HuBERT+KM (Lakhotia et al., 2021) | | | 5.8±0.2 | 14.6±0.1 | 6.0±0.1 | 14.9±0.1 |
| dMel | | | 6.0±0.4 | 15.2±0.8 | 6.1±0.4 | 15.7±0.7 |
| SpeechTokenizer (Zhang et al., 2023b) | Base | LS-960 | 6.5±0.3 | 16.9±0.7 | 6.9±0.4 | 17.5±0.5 |
| HuBERT+KM (Lakhotia et al., 2021) | | | 5.3±0.1 | 13.7±0.2 | 5.8±0.1 | 13.8±0.1 |
| dMel | | | **3.8**±0.1 | **10.3**±0.1 | **4.2**±0.2 | **10.4**±0.1 |

Table 15: Ablations on the LM-style ASR model with GPT-2 architecture using the setup from Chen et al. (2024): we ablate pretraining with text, speed perturbation and speech tokenization methods. All models are trained on LibriSpeech 960h. We report WER (%) on LibriSpeech validation and test sets.

| Tokenization | Text Pretraining | Speed Perturbation | dev-clean↓ | dev-other↓ | test-clean↓ | test-other↓ |
|---|---|---|---|---|---|---|
| BPE on HuBERT+KM, Chen et al. (2024) | ✓ | ✓ | 2.9 | 6.2 | 3.0 | 6.6 |
| BPE on HuBERT+KM, reproduction | ✓ | ✓ | 2.9 | 6.3 | 3.2 | 6.7 |
| BPE on HuBERT+KM | ✗ | ✓ | 3.6 | 7.8 | 3.8 | 8.3 |
| HuBERT+KM | ✗ | ✓ | 5.1 | 8.9 | 5.5 | 9.3 |
| dMel | ✗ | ✓ | 3.1 | 8.4 | 3.4 | 8.6 |
| BPE on HuBERT+KM | ✗ | ✗ | 5.4 | 9.7 | 5.3 | 10.0 |
| HuBERT+KM | ✗ | ✗ | 6.0 | 10.5 | 6.4 | 10.9 |
| dMel | ✗ | ✗ | 3.7 | 9.7 | 3.9 | 9.5 |

## F.3 JOINT SPEECH-TEXT MODELING DISCUSSION

We found it to be challenging to train joint model for ASR and TTS, similar to observations as in Maiti et al. (2024) and e.g. Shi et al. (2022); Rouditchenko et al. (2024) for joint audio-visual speech recognition. Also, there is a very recent research work (Toyin, 2024), that also shows training TTS and ASR jointly is challenging, and needs carefully designed model architecture and training loss fusion technique.

One of the reasons is the different pace of learning. Careful consideration of training strategies can mitigate some of the challenges in joint modeling of TTS and ASR tasks, highlighting the complexities inherent in combining these distinct but related tasks within a single model.

Another reason we suspect is the mismatch between train and test time, which is more pronounced for the joint modeling: if we compare individual validation losses per task in joint model to their one-task training counterparts we see they match each other (so training is fine), however the generation (test time which mismatches how the train loss is defined) for both tasks is broken: longer sequences has hallucination and high repetition issues. This could be due to different length of sequences between text and audio and thus learnt attention pattern could be different which creates longer sequences generation issue for the joint model.

Last but not the least, the two tasks have opposite modalities in the input and output, making it rather difficult to model. Most previously researched multi-task work have the same modality in the output. The combination of ASR and TTS is a rather recent phenomenon, such as Viola and VOXTLM.

