# OpenReview forum: "dMel: Speech Tokenization Made Simple"
_ICLR.cc/2025/Conference — Submitted to ICLR 2025_

### Official Review · Reviewer_wr2J · 2024-10-18

**Soundness:** 3
**Presentation:** 3
**Contribution:** 3
**Rating:** 6
**Confidence:** 4

**Summary:**

This work propose to solve the problem of codec: it is hard for one codebook to cover both semantic and acoustic information, but multiple codebook will complicate the architecture and require additional pretraining. Therefore, this work proposes to discretize mel-filterbank channels into discrete intensity bins, which produces a simple representation that outperforms existing speech
tokenization methods.

I believe this is a pioneering work that may open a potentially new track of TTS research --> use continous mel to replace discrete codec in lm base TTS.

One question:
- How is it compared to another similar work MELL-E (https://arxiv.org/abs/2407.08551) that also use continous mel tokens for lm based TTS?

**Strengths:**

see above

**Weaknesses:**

see above

**Questions:**

see above

---

> ### Author Response · Authors · 2024-11-17
>
> We sincerely appreciate your insightful review and for recognizing this as `pioneering work that may open a potentially new track of TTS research.` Your understanding of our core motivation - addressing the challenge of capturing both semantic and acoustic information without architectural complexity - perfectly aligns with our research goals.
>
> We appreciate the reviewer bringing MELL-E (https://arxiv.org/abs/2407.08551) to our attention. However, we would like to clarify two important points:
>
> * MELL-E was released on July 11, 2024. According to ICLR 2025's policy, this qualifies as concurrent work, as it was published during our paper's preparation period. However, the concurrent emergence of similar ideas from different teams often suggests the research direction is promising
> * While both works address speech representation, MELL-E employs a more complex architectural approach and operates on continuous mel features. Our method operates on discrete features (opposite direction to continuous features) and prioritizes simplicity and efficiency while achieving competitive performance. Also, our work demonstrates broader applicability across both ASR and TTS tasks
>
> We believe these concurrent developments validate the importance of exploring mel-based representations for language model-based speech processing.

---

> > ### Comment · Reviewer_wr2J · 2024-11-26
> > **keep my rating**
> >
> > Thanks for response, i would like to keep my postive rating

---

### Official Review · Reviewer_Wgkg · 2024-10-21

**Soundness:** 3
**Presentation:** 2
**Contribution:** 3
**Rating:** 5
**Confidence:** 4

**Summary:**

This work propose to discretize mel-spectrum into a special kind of intensity bins, which is proved to be a simple representation but more effective than commonly used speech tokenizers (i.e., codec). The authors claim that the newly proposed dMel well carry both acoustic and semantic information within speech signal, without losing information during quantization like codec. Experimental results have proved the effectiveness in tts and asr tasks.

**Strengths:**

- New idea of using mel spectrum, which is continuous signal, for language modeling. This is different from recently popular codec based TTS.

**Weaknesses:**

- For TTS and ASR evaluation, there are only limited baselines for comparison, more powerful models like vall-e (TTS) and whisper (ASR) should also be included.
- The results of dMel are only reported on top of RichTTS and RichASR, experiments on more backbones are expected for better evaluation.

**Questions:**

- For RichTTS and RichASR, what about the implementation details like architecture/training data (compare to speechgpt?)
- Is there any open-source plan to support the community?

---

> ### Author Response · Authors · 2024-11-17
>
> We sincerely appreciate the reviewer noting our work's soundness and contribution, particularly in recognizing the novelty of our mel-spectrum approach. We've carefully considered your feedback and would like to address each point comprehensively:
>
>
> ## 1. Baseline Comparisons
> While we understand the interest in Whisper/VALL-E comparisons, there are fundamental methodological reasons these were not included:
>
>   * Neither is open-source, preventing reproducible research. Whisper uses proprietary, undisclosed training data.
>   * Such comparisons wouldn't validate our core scientific contribution
>   * Our focus is on advancing open, reproducible speech research
>
> ## 2. On Architecture Experiments:
>
> We apologize if Table 10's extensive architecture experiments weren't sufficiently highlighted, but we reported results across multiple standard architectures in Table 10:
>    * Transformer Decoder (most popular architecture)
>    * CTC-based models
>    * Sequence-to-sequence (encoder-decoder) models (whisper falls into this type of models, but was trained on ~400x more data, so we actually compare with the family of models whisper is based on)
>
> Our experiments demonstrate dMel's effectiveness across multiple standard architectures in a reproducible setting with open data.
>
> ## 3. Response to questions
>
> > For RichTTS and RichASR, what about the implementation details like architecture/training data (compare to speechgpt?)
>
> The architecture is introduced in Table12. The training data is LibriSpeech for most of the Tables except Table 5 and 6, as we introduced in Section 3.1. Table 5 and 6 are using different datasets to compare the results with different models.
>
> > Is there any open-source plan to support the community?
>
> We will definitely open-source our full codebase and we are undergoing necessary steps to release full code.

---

> ### Author Response · Authors · 2024-12-01
> **Reminder of rebuttal response**
>
> Thank you once again for your thoughtful review and feedback! As we approach the end of the discussion period, we want to ensure that our previous responses have fully addressed all your concerns. If you have any additional questions or unresolved issues that we can clarify, please don’t hesitate to let us know. We’re more than happy to assist further!

---

### Official Review · Reviewer_wcjv · 2024-10-21

**Soundness:** 2
**Presentation:** 2
**Contribution:** 2
**Rating:** 3
**Confidence:** 5

**Summary:**

This paper proposes dMel, a simple method for quantizing Mel spectrograms into discrete units for LM-style decoder-only ASR and TTS. Unlike self-supervised semantic tokens and neural codecs, dMel is parameter- and optimization-free. Experimental results indicate superior ASR and TTS performance compared to prior methods like HuBERT + K-means and SpeechTokenizer.

**Strengths:**

The proposed dMel mitigates the issues in existing speech tokenizers. First, prior works like self-supervised learning (SSL) based tokenizers require extensive pre-training and sometimes not being able to preserve acoustic details for speech generation and synthesis. Second, neural codecs preserve fine-grained acoustic representations but might not be able to perform ASR and TTS because of the weak correlations between codebooks and frames. The authors propose a parameter- and training-free approach to achieve similar ASR and TTS performance.

**Weaknesses:**

Despite the success of the dMel method presented in the experiment results, the following issues question its novelty and effectiveness.

1) **Bitrate:**
Bitrate is a crucial metric for comparing different tokenizers in prior studies but is not included in this paper. According to the provided information, dMel@40Hz, HuBERT-KM, and SpeechTokenizer, respectively, have bitrates of 12.8, 0.4, and 4kbps. The huge difference in bitrates might lead to an **unfair comparison**. Moreover, the number of centroids of K-means clustering in HuBERT-KM could be increased since 200 is considered a small codebook size (Table 2), while 500 and larger values are more commonly used in past literature.

2) **Baselines:**
Advances in speech tokenization techniques have improved many downstream applications, including ASR and TTS. However, this paper only compares dMel with HuBERT + K-means and SpeechTokenizer, where the K-means method was proposed in 2021 [1]. Also, speech tokenization papers usually consider spoken language modeling a standard evaluation task [2,3,4].

3) **Writing:**
Writing could be improved with the assistance of writing tools, including LLMs. For instance, from lines 299 to 301, the original text is "From Table 3, we can see that semantic tokenization (HuBERT-KM) is not good for speech reconstruction. Meanwhile, acoustic tokenizers that are optimized to reconstruct the signal directly (EnCodec and SpeechTokenizer) do well." The sentence is generally clear, but in academic writing, it's often better to use more precise language and avoid subjective terms like "not good" or "do well." A revised version is "Table 3 shows that semantic tokenization (HuBERT-KM) performs poorly in speech reconstruction, while acoustic tokenizers optimized for direct signal reconstruction (EnCodec and SpeechTokenizer) demonstrate superior performance."

[1] Lakhotia, Kushal, et al. "On generative spoken language modeling from raw audio." Transactions of the Association for Computational Linguistics 9 (2021): 1336-1354.
[2] Gat, Itai, et al. "Augmentation invariant discrete representation for generative spoken language modeling." arXiv preprint arXiv:2209.15483 (2022).
[3] Borsos, Zalán, et al. "Audiolm: a language modeling approach to audio generation." arXiv preprint arXiv:2209.03143 (2022).
[4] Défossez, Alexandre, et al. "Moshi: a speech-text foundation model for real-time dialogue." arXiv preprint arXiv:2410.00037 (2024).

Note: dMel@40Hz bitrate = $40 \times 80 \times \log_2 16 = 12800 = 12.8$kbps

**Questions:**

1) What are the hyperparameters for extracting log Mel spectrograms? Window size? Stride?
2) Why are the model names "RichASR" and "RichTTS?" Any specific reasons?
3) What is the codebook utilization rate or distribution of dMel? The proposed quantization approach divides the intensity into equally-spaced bins. However, a potentially better way is to assign bin sizes according to the data distribution for a uniform codebook utilization.
4) Does pre-training the LM with speech-only data help downstream performance? In spoken LM applications, it is common to pre-train the LM on speech tokens with large unlabeled data.
5) Are there any decoding techniques involved in RichTTS and RichASR? E.g., beam search.

---

> ### Author Response · Authors · 2024-11-17
> **Response to Reviewer wcjv 1/2**
>
> Thank you for your detailed review. We address your concerns and questions below:
>
> ## 1. Regarding Bit Rate and Compression
>
> We cannot agree on different bit-rate leading to `unfair comparison` as bit-rates don't necessarily correlate with better downstream task performance. Several work have observed this:
> * Moshi [1] observed: "Across our experiments, we make the somehow counter-intuitive observation that this gain gets more significant as we lower the bitrate."
>  * DASB [2] similarly reported: "Interestingly, higher bitrates, when available (e.g., for EnCodec and DAC), tend to degrade performance."
>  * Another evidence is in VoxtLM [3] paper, they found using k=200 centroids achieved 3.5 CER for TTS but using k=1000 centroids resulted in a higher 6.1 CER for TTS. This also indicates higher bit-rate doesn’t mean better results.
>
> 2. It's important to note that traditional bit-rate comparisons may not be directly applicable to dMel due to its distinct architectural features:
>
>     * Encoder-free Design: Unlike conventional approaches, dMel operates without an audio compression model, making traditional compression rate metrics less relevant and inaccurate to measure dMel.
>     * Parallel Processing: While dMel utilizes 80 channels per frame, these channels are processed in parallel during both encoding and decoding. Therefore:
>         * Computational complexity is primarily determined by frame rate rather than bit-rate
>         * Traditional bit-rate calculations do not accurately reflect the model's efficiency
>
> We respectfully suggest that dMel's performance should be evaluated within the context of its novel modeling approach rather than compression method, where frame rate serves as a more meaningful metric than bit-rate.
> We would like to add discussion about the bit-rate and frame-rate in our paper too.
>
>
> 3. Choice of Centroids for HuBERT K-means
>
> Regarding the number of centroids used in our HuBERT k-means implementation, our choice of 200 centroids follows the methodology established in VoxTLM[3], which demonstrated superior performance with this configuration. Specifically, VoxTLM's ablation studies (Table 6) show that:
>
> * k=200 centroids achieved 3.5 CER for TTS
> * k=1000 centroids resulted in a higher 6.1 CER for TTS
>
> This empirical evidence supports our choice of centroid count.
>
> ## 2. Regarding Baseline Comparisons
>
> We appreciate the reviewer's comments about baselines, but several points need clarification:
>
> * Regarding baseline selection:
>     * SpeechTokenizer is a very recent work (ICLR 2024)
>     * Even Moshi [1], which you cited, builds upon SpeechTokenizer's methodology, acknowledging "inspiration from previous work on SpeechTokenizer"
> * Regarding Spoken Language Modeling (SLM): The assertion that SLM is a "standard evaluation task" for speech tokenization papers is not accurate. Many significant works do not include SLM evaluation yet:
>     * EnCodec
>     * SpeechTokenizer
>     * Recent works mentioned by reviewer 6UMS (SPECTRAL CODECS, SemantiCodec, APCodec, Single-Codec)
> * Regarding cited papers:
>     * [4] (AudioLM) doesn't propose new tokenization methods but focuses on modeling existing tokens
>     * [2] focuses on representation robustness for SLM rather than tokenization
>     * [1] (Moshi) was released just one week before submission deadline and it is a 67 pages paper works on both speech tokenization and SLM.
>
> ## 3. Regarding Writing
>
> Thanks for spotting places with less formal style in our paper. We will correct accordingly the style, so please let us know if you found any other places. At the same time, we are concerned about feedback on usage LLMs for writing as this could violate the privacy.
>
> ```
> [1] Moshi: a speech-text foundation model for real-time dialogue
> [2] DASB - Discrete Audio and Speech Benchmark
> [3] Voxtlm: unified decoder-only models for consolidating speech recognition/synthesis and speech/text continuation tasks.
> [4] Borsos, Zalán, et al. Audiolm: a language modeling approach to audio generation.
>
> ```

---

> ### Author Response · Authors · 2024-11-17
> **Response to Reviewer wcjv 2/2**
>
> ##  Response to the Questions:
>
> > What are the hyperparameters for extracting log Mel spectrograms? Window size? Stride?
>
> The stride size is 1000/frame_rate. For our 40HZ feature, the stride size is 25ms, the window size is 50ms. We have introduced these in the caption of Table 10.
>
> > Why are the model names "RichASR" and "RichTTS?" Any specific reasons?
>
> We named our model informally because of internal reasons, not because of any scientific reason - sorry if that caused confusion.
> > What is the codebook utilization rate or distribution of dMel? The proposed quantization approach divides the intensity into equally-spaced bins. However, a potentially better way is to assign bin sizes according to the data distribution for a uniform codebook utilization.
>
> We ablated dMel uniform binning with a percentile-based discretization method, which computes bin boundaries based on channel-specific statistics from the LibriSpeech training data. The latter showed competitive but slightly inferior performance compared to our proposed method and we have left this for future exploration.
> > Does pre-training the LM with speech-only data help downstream performance? In spoken LM applications, it is common to pre-train the LM on speech tokens with large unlabeled data.
>
> While speech-only pre-training could be valuable future work, our current focus is specifically on:
>   * Demonstrating effective mel feature discretization
>   * Establishing a decoder-only architecture for conditional sequence generation, where speech is generated from text input (TTS) or text is generated from speech input (ASR)
>
> Our strong ASR/TTS results demonstrate these core contributions are effective without additional pre-training. Future work could explore both pre-training for multi-task scenarios and unconditioned generation tasks.
>
> > Are there any decoding techniques involved in RichTTS and RichASR? E.g., beam search.
>
> Thanks for pointing out. We use top-p (p=0.95) sampling instead of beam search for RichTTS while no beam-search is used for RichASR and simple greedy decoding is done. We will add this important implementation details in our manuscript.

---

> > ### Comment · Reviewer_wcjv · 2024-11-23
> >
> > I appreciate the authors' clarifications.
> >
> > First, I agree that a higher bitrate does not necessarily imply better performance. However, many prior works related to audio codec and speech tokenization, like WavTokenizer [1] and SemanticCodec [2], try to achieve ultra-low bitrate discretization. In contrast, dMel has a significantly higher bitrate compared with those previous studies, which deviates from the current trend of research. Besides the usefulness of the demonstrated tasks (ASR and TTS), I do not see other practical usages of dMel compared with other data-driven speech tokenizers.
> >
> > Moreover, it is unclear whether dMel is useful for joint speech recognition and generation (Table 11). Since VOXTLM operates on top of HuBERT units and performs well on ASR and TTS, these units may be more suitable for decoder-only speech LMs.
> >
> > Another unaddressed issue is the robustness of dMel. Data-driven methods like self-supervised learning and neural codecs could be easily improved by introducing more diverse data to increase robustness [3,4]. However, it is unknown whether noise and perturbation affect dMel.
> >
> > Hence, I intend to keep the score unchanged.
> >
> > ---
> >
> > [1] Ji, Shengpeng, et al. "Wavtokenizer: an efficient acoustic discrete codec tokenizer for audio language modeling." arXiv preprint arXiv:2408.16532 (2024).
> > [2] Liu, Haohe, et al. "SemantiCodec: An Ultra Low Bitrate Semantic Audio Codec for General Sound." arXiv preprint arXiv:2405.00233 (2024).
> > [3] Gat, Itai, et al. "Augmentation invariant discrete representation for generative spoken language modeling." arXiv preprint arXiv:2209.15483 (2022).
> > [4] Messica, Shoval, and Yossi Adi. "NAST: Noise Aware Speech Tokenization for Speech Language Models." arXiv preprint arXiv:2406.11037 (2024).

---

> > > ### Author Response · Authors · 2024-11-23
> > > **Response - 3**
> > >
> > > We appreciate your acknowledgment that “a higher bitrate does not necessarily imply better performances”. This directly addresses one of your initial rejection reasons about "unfair comparison" due to bitrate differences. We also note that you haven't responded to our clarification about the baseline concern, where we explained that SpeechTokenizer (ICLR 2024) represents a very recent and strong baseline in the field.
> > >
> > > In this response, we would like to address your three new concerns that appear to motivate the rejection decision:
> > >
> > >
> > > > 1. dMel has higher bit-rate, while many existing works are pursuing low bit-rate, which  deviates from the current trend of research.
> > >
> > >
> > > First, attempts to achieve ultra-low bitrate do not imply that this is the right direction for all research. Scientific progress often comes from exploring alternative approaches, and higher bit rates may offer valuable trade-offs worth investigating.
> > >
> > > Second, the current trend focuses on compression-based tokenization, where compression rate directly affects token rate for Transformer modeling. However, dMel fundamentally differs in its architecture - it operates without a compression model, and our token rate is clearly presented as frame rate in Table 3.
> > >
> > >
> > >
> > > > 2. Moreover, it is unclear whether dMel is useful for joint speech recognition and generation (Table 11). Since VOXTLM operates on top of HuBERT units and performs well on ASR and TTS, these units may be more suitable for decoder-only speech LMs.
> > >
> > >
> > > We must respectfully point out a factual error in the statement "VOXTLM performs well on ASR and TTS". In fact, VOXTLM demonstrates that HuBERT tokens yield poor TTS performance. Our Table 11 shows that dMel maintains strong TTS performance while achieving meaningful ASR results.
> > >
> > > In fact, no existing tokenization method has demonstrated SOTA performance for both tasks simultaneously.
> > > Recent works [1,2] explicitly highlight the challenges in balancing ASR and TTS performance in a single model. Notably, [2] (EMNLP 2024) achieves superior results through dedicated model design rather than tokenization innovation, using mel-spectrogram instead of codec features. These findings provide no evidence that HuBERT or other units are more suitable than dMel for speech-text models.
> > >
> > >
> > >
> > > > 3. Another unaddressed issue is the robustness of dMel. Data-driven methods like self-supervised learning and neural codecs could be easily improved by introducing more diverse data to increase robustness [3,4]. However, it is unknown whether noise and perturbation affect dMel.
> > >
> > >
> > > The reviewer's concern about robustness actually highlights a key advantage of dMel. While data-driven methods can be improved by introducing more diverse data, they inherently suffer from information loss due to their neural compression nature - they learn to discard information based on training data, which may prove crucial in unseen conditions. In contrast, mel-spectrogram is a physics-based signal representation that:
> > >
> > >   1. Preserves frequency components through principled transformation - while it does not retain phase information, this aligns with human auditory perception which is primarily sensitive to magnitude spectrum
> > >   2. Has demonstrated robust performance over decades of speech processing research
> > >   3. Does not require data-dependent training to handle different acoustic conditions
> > >   4. Maintains consistent behavior across various noise conditions due to its deterministic nature
> > >
> > > This fundamental difference means that while data-driven methods need to be explicitly trained to handle noise and perturbations (potentially missing unknown variations), dMel's robustness is inherent in its signal processing foundation. The reviewer's suggestion of "improving robustness through more diverse data" actually underscores the limitations of data-driven approaches - they need to "learn" what mel-spectrogram already captures by design.
> > >
> > >
> > > We sincerely appreciate your thoughtful review and the opportunity to clarify these points. Your feedback helped us better articulate our method's unique contributions and position in the field. We believe the above clarifications address the core concerns about our work's direction, effectiveness, and robustness. We hope this discussion has been valuable in highlighting dMel's distinct advantages and contributions to the field.
> > >
> > > ```
> > > [1] [A Unified Model for Text-to-Speech and Speech-to-Text.](https://dclibrary.mbzuai.ac.ae/mletd/32/)
> > > [2][STTATTS: Unified Speech-To-Text And Text-To-Speech Model] (https://aclanthology.org/2024.findings-emnlp.401.pdf)
> > > ```

---

### Official Review · Reviewer_qtqg · 2024-11-03

**Soundness:** 3
**Presentation:** 3
**Contribution:** 2
**Rating:** 5
**Confidence:** 4

**Summary:**

This paper presents dMel, an encoder-free speech tokenizer that simplifies speech tokenization by discretizing log mel-filterbank outputs into discrete intensity bins, eliminating the need for complex encoding architectures. Unlike previous tokenization methods that separate semantic and acoustic information, dMel maintains both in a single, unified representation. The tokenization process reduces precision of each filter output per frame while retaining the essential information needed for high-quality speech resynthesis, achieved by leveraging pre-trained vocoders. Additionally, the paper explores the application of dMel in language model (LM)-style training for both automatic speech recognition (ASR) and text-to-speech (TTS) tasks. The results demonstrate that dMel performs comparably or better than existing methods in preserving semantic content and reconstructing natural-sounding audio. This efficient, unified approach to speech tokenization facilitates streamlined ASR and TTS training, advancing joint modeling of speech and text.

**Strengths:**

- The idea of quantizing mel spectrogram as tokenization is interesting and simple (in a good way).
- Results on TTS and ASR show dMel quantization has a small impact on models trained on continuous representation, training downstream models on top of dMel also provided similar results to their continuous counterparts.  These observations are interesting, showcasing the generalizability of dMel.
- Overall, I believe dMel is much more efficient in terms of model size and inference speed comparing to existing speech tokenizers (but this part is not well evaluated in the experiment section, see weaknesses).

**Weaknesses:**

- As a speech tokenization paper, this work lacks a discussion on the overall bit rate for compression besides frame rate. Especially in the comparison with the prior works (e.g., Table 3).  dMel is over 12.8kbps~5kbps (assuming 40 fps $\times$ 32 mel filters $\times$ 4 bit-per-filter)~, which is higher than Hubert-KM and Speech Tokenizer.

- This paper spent most of the space discussing ASR & TTS systems based on dMel. While the numbers are good, it is still not as good as a normal mel spectrogram (which is expected). This makes the content of the paper somewhat sparse, which is the biggest weakness in my opinion. The current paper seems to only suggest dMel is a spectrogram quantization approach, as it is essentially lowering numerical precision and showing the distortion is minimal on vocoder, ASR, and TTS. It would be more interesting to involve some other studies, for example:
  - Efficiency-related studies, such as how the encoder-free and lightweight-decoder design of dMel can speed up or lower memory usage downstream applications.
  - Applications where speech tokenization matters more, e.g., spoken LM [1,2], would better justify whether dMel can be viewed as a good speech tokenization approach.

These scenarios/experiments would all be more suitable (than just plain ASR/TTS WER/MOS) for assessing the value of dMel. (I would like to note that these are not concerns that are expected to be addressed during the ICLR rebuttal period, they can be viewed as suggestions for future version of the paper)


[1] https://arxiv.org/pdf/2102.01192

[2] https://arxiv.org/abs/2410.00037

**Questions:**

- Is there a fundamental difference between finite scalar quantization (FSQ; [1]) and dMel's quantization?
If not, I think the FSQ paper should be acknowledged.

[1] https://arxiv.org/pdf/2309.15505

---

> ### Author Response · Authors · 2024-11-17
>
> We sincerely appreciate your thorough review and recognition of our novel contributions. Below we address your key concerns:
>
> ## 1. Regarding Bit Rate and Compression
>
> You raise an important point about bit rate discussion. While dMel operates at approximately 12.8 kbps (40fps × 80 mel filters × 4 bits/filter), we'd like to highlight several important considerations:
>
> 1. Recent research has demonstrated that higher bit rates don't necessarily correlate with better downstream task performance. For instance:
>
>     * Moshi [1] observed: "Across our experiments, we make the somehow counter-intuitive observation that this gain gets more significant as we lower the bitrate."
>     * DASB [2] similarly reported: "Interestingly, higher bitrates, when available (e.g., for EnCodec and DAC), tend to degrade performance."
>
> 1. It's important to note that traditional bit-rate comparisons may not be directly applicable to dMel due to its distinct architectural features:
>     * Encoder-free Design: Unlike conventional approaches, dMel operates without an audio compression model, making traditional compression rate metrics less relevant and inaccurate to measure dMel, since the encoders participate in the compression scheme of the other models.
>     * Parallel Processing: While dMel utilizes 80 channels per frame, these channels are processed in parallel during both encoding and decoding. Therefore:
>         1. Computational complexity is primarily determined by frame rate rather than bit-rate
>         2. Traditional bit-rate calculations do not accurately reflect the model's efficiency
>
> We respectfully suggest that dMel's performance should be evaluated within the context of its novel modeling approach rather than compression method, where frame rate serves as a more meaningful metric than bit-rate.
> We would like to add discussion about the bit-rate and frame-rate in our paper too.
>
> ## 2. Regarding Content and Contributions
>
> We appreciate your feedback about the paper's focus on ASR & TTS systems. We would like to emphasize three key aspects that demonstrate the broader impact of our work:
>
> * Novel Architecture: While mel features have been extensively studied, our work is the first to investigate modeling them with a decoder-only architecture. The dMel + decoder combination represents a fundamental architectural innovation in the field.
> * Superior TTS Performance: Contrary to the expectation that our approach might underperform traditional mel spectrograms, Table 5 demonstrates that RichTTS (dMel + TransformerDecoder) achieves lower WER compared to popular open-source mel-based TTS models including VITS, FastSpeech2, and Tacotron2.
> * Innovative Applications: By aligning TTS model design with Language Model architectures, dMel enables the application of numerous LM techniques to speech synthesis, including:
>     * Speculative decoding
>     * KV-caching
>     * Multi-turn capabilities
>     * And potentially many more
>
> We will revise the manuscript to better emphasize these broader implications and their potential impact on the field.
>
> [1] Moshi: a speech-text foundation model for real-time dialogue  https://arxiv.org/abs/2410.00037
>
> [2] DASB - Discrete Audio and Speech Benchmark https://arxiv.org/abs/2406.14294
>
>
> ## 3. Questions
> >Is there a fundamental difference between finite scalar quantization (FSQ; [1]) and dMel's quantization? If not, I think the FSQ paper should be acknowledged.
>
>
> We recognize that our quantization method shares some similarities with FSQ, but also differs in key aspects:
>
>  1) FSQ employs scalar quantization in a learned latent code space, primarily focusing on image data modeling. In contrast, our work pioneers the successful application of scalar quantization technique to the original raw mel-frequency band (mel-fb) space, serving as a training-free speech tokenization approach.
>  2) FSQ necessitates an additional bound operation to limit the range of the latent codes. Moreover, the dimensionality of the code in FSQ is considerably smaller (≤6) compared to ours (80). This higher dimensionality is crucial for maintaining speech quality across a broader frequency range.
>
> We will make sure to cite the FSQ paper appropriately in our revised manuscript to acknowledge this prior work. Our primary contribution is not the quantization method itself, but rather:
>
> * Successfully applying it to mel-frequency features for speech discretization
> * Developing an integrated architecture that enables decoder-only speech modeling
> * Demonstrating strong empirical results across multiple downstream tasks

---

> > ### Comment · Reviewer_qtqg · 2024-11-24
> >
> > I would like to thank the authors for their response.
> >
> > > Recent research has demonstrated that higher bit rates don't necessarily correlate with better downstream task performance.
> >
> > Yes, but my concern is that dMel is having **higher** bit rate than the baselines it's being compared against.
> >
> > > We respectfully suggest that dMel's performance should be evaluated within the context of its novel modeling approach rather than compression method, where frame rate serves as a more meaningful metric than bit-rate. We would like to add discussion about the bit-rate and frame-rate in our paper too.
> >
> > While I can understand the claim that frame rate is more important than bit rate for dMel, I don't fully agree with the author
> > - Frame rate does not reflect the number of bins used for quantization. The precision of quantization obviously matters (as shown in Table 9).
> > - For downstream applications, bit rate is at least equally as frame rate, since it directly impacts the architecture, complexity, and learning dynamics (e.g., number of prediction heads for the token, loss components, etc.) of the downstream model.
> > - At the very least, I believe bitrate should be disclosed explicitly when compared to prior works on speech reconstruction (as in those audio codec papers cited by this work) and discussed as a limitation or in the paper.

---

> > > ### Author Response · Authors · 2024-11-25
> > > **Response to Reviewer qtqg - 2**
> > >
> > > We appreciate your timely response and constructive feedback regarding the bit rate discussion. You raise an important concern over complexity and we would like to clarify how dMel's higher bit rate impacts practical performance, hoping to address your concerns.
> > >
> > > While bit rate is important for reconstruction fidelity, the computational complexity of handling tokenization is more critical. This involves three aspects:
> > >
> > > * Audio-to-Token Complexity: dMel is parameter-free, unlike low bitrate tokenizers that require trained models (Table 3)
> > > * Token Modeling Complexity: dMel only requires a reshaping operation and an optional linear transformation layer (we detailed this below), with softmax over a small vocabulary of 16 tokens. In contrast, low-bit rate tokenizers typically require both autoregressive and non-autoregressive models for primary codes and residual codes, plus larger vocabularies that significantly increase embedding parameters and softmax computations.
> > > * Token-to-Audio Complexity: Our TTS experiments utilize a lightweight 1M parameter open-source vocoder, leveraging years of research in mel-to-audio conversion.
> > >
> > >
> > >
> > > > Frame rate does not reflect the number of bins used for quantization. The precision of quantization obviously matters (as shown in Table 9).
> > >
> > >
> > > Yes, we agree it matters for reconstruction fidelity. However, Table 9 demonstrates that increasing bins can actually worsen TTS error rates.
> > >
> > >
> > > > For downstream applications, bit rate is at least equally as frame rate, since it directly impacts the architecture, complexity, and learning dynamics (e.g., number of prediction heads for the token, loss components, etc.) of the downstream model.
> > >
> > >
> > >
> > > Yes, we partially agree: considering high bitrate for dmel, this only leads to marginal increase in complexity.
> > >
> > > In fact, our model didn’t increase the number of prediction heads or changing the loss components, but with standard prediction head and cross-entropy loss operations with one additional dimension. We are not sure if this concern is from assuming our model needs 80*16 embeddings or not. But we think it is important to restate:
> > >
> > > * All 80 channels share the same vocabulary embeddings (16 total embeddings, not 80*16. Single prediction head, not 80 heads)
> > >
> > > Implementation requires only:
> > >
> > > * Reshaping operation from (80*x) → (80, x) (this is the last layer’s hidden states tensor of Transformer decoder)
> > > * Softmax over the reshaped tensor
> > > * Standard cross entropy loss operations with one additional dimension
> > > * Optional linear layer when hidden states aren't divisible by 80
> > >
> > > So, a reshaping operation and an optional linear operation is the cost difference compared to vanilla Transformer. This is simpler than low bit-rate compression tokens requiring separate models for different code components (coarse-codec autoregressive and fine-codec non-autoregressive transformers).
> > >
> > > Frame rate remains the primary complexity driver, hence our emphasis on evaluating dMel in the frame rate context.
> > >
> > >
> > > > At the very least, I believe bitrate should be disclosed explicitly when compared to prior works on speech reconstruction (as in those audio codec papers cited by this work) and discussed as a limitation or in the paper.
> > >
> > >
> > > Yes, we agree with your point. Through the discussion here, we admit we need to clarify the bit-rate of dMel and how it affects the comparison and downstream applications.

---

> > > > ### Comment · Reviewer_qtqg · 2024-11-29
> > > >
> > > > I would like to thank the authors for their further clarification.
> > > >
> > > > > Yes, we agree it matters for reconstruction fidelity. However, Table 9 demonstrates that increasing bins can actually worsen TTS error rates.
> > > >
> > > > To clarify, my point is **bitrate does matter for speech tokenization methods, and it should be preferred over frame rate for fair comparison**. This is to contradict the authors' original claim that *"frame rate serves as a more meaningful metric than bit-rate"*. The fact that dMel with 8 bins (9.6kbps) is significantly worse than 16 bins is strong evidence.
> > > >
> > > > **Comparing dMel against other methods by only showing the frame rate is unfair**. In fact, if we add bitrate to Table 3, it immediately reveals that dMel is only slightly better (or even worse) than SpeechTokenizer, while the latter is at a significantly lower bitrate. It is unfair to claim that dMel *``performs better than other existing speech tokenization methods''* for this very reason.
> > > >
> > > > As for the observation *``increasing bins can actually worsen TTS error rates"*, it is not something that is relevant to comparing fairly against other methods, but more of dMel's own special property.
> > > >
> > > > For downstream applications, I am satisfied with the authors' responses.
> > > >
> > > > While dMel has its own value in other aspects (e.g., training-free encoder, less dependencies between codes, etc.), I believe some of the key arguments in this paper are invalid, and some of the comparisons against prior works are unfair as pointed out above. Hence, I would like to keep my initial rating.
> > > >
> > > > p.s. In Table 2, bit-rate should be bitrate, and kps should be kbps.

---

> > > > > ### Author Response · Authors · 2024-11-30
> > > > >
> > > > > Thank you for your detailed feedback. We appreciate your patience and would like to provide further clarification regarding our results and claims.
> > > > >
> > > > > Regarding Table 3, we want to emphasize that our intention was not to claim superiority in speech reconstruction or to present dMel as the state-of-the-art method. Instead, the table serves to demonstrate dMel's fundamental characteristics in preserving speech content. Our results show that dMel achieves comparable reconstruction quality to mel-spectrograms and existing methods, which was our primary objective for this comparison. Also, Table 3 serves a crucial purpose: it quantifies dMel's information retention relative to mel-spectrograms, which is essential since mel-spectrograms cannot be directly used for decoder-only downstream tasks. So we have to use Table 3 to measure the difference. In fact, in Table 3, these results provide no basis for claiming dMel's superiority in reconstruction quality:
> > > > > - EnCodec has the lowest WER while dMel's WER is close to it.
> > > > > - Mel-HifiGAN and dMel-HifiGAN has better MOS-LQO.
> > > > > - SpeechTokenizer leads in MOS scores, while dMel-HifiGAN performing similar to EnCodec.
> > > > >
> > > > >
> > > > > Our broader claim about dMel's superior performance specifically refers to downstream tasks, not reconstruction quality. The primary strength of dMel lies in its effectiveness as a foundation for generation model such as text to speech generation and speech to text generation. This distinction is important, as our focus is on dMel's utility in these downstream applications rather than pure reconstruction performance.
> > > > >
> > > > > We have reviewed our manuscript carefully and believe we have not made such claims about reconstruction. However, we acknowledge the importance of clearly stating both the limitations of dMel in reconstruction and the specific purpose of Table 3 in our analysis. We will add this in our manuscripts and we hope these clarifications help present our work's contributions more accurately and address the reviewer's concerns regarding our claims.

---

### Official Review · Reviewer_6UMS · 2024-11-04

**Soundness:** 3
**Presentation:** 3
**Contribution:** 3
**Rating:** 8
**Confidence:** 4

**Summary:**

The paper introduces a novel approach to speech tokenization by discretizing mel-filterbank channels. This method effectively preserves both semantic and acoustic information, offering an interpretable, model-free representation grounded in the raw acoustic space. The authors train a transformer-based language model for speech-text modeling and evaluate their proposed tokenization approach on speech recognition (ASR) and speech synthesis (TTS) tasks.

**Strengths:**

- The proposed method is efficient, as it avoids hierarchical dependencies among mel-spectrogram channels, allowing for independent modeling of each channel within each frame using a straightforward, decoder-only (LM-style) transformer architecture.
- The approach is robust, simple yet innovative, with comprehensive evaluations that support the design choices.
- The encoder operates independently of the decoder, unlike many other tokenizers, making it compatible with any vocoder that accepts mel-spectrogram inputs.
- A detailed analysis of the setup is provided to enhance reproducibility.
- The paper is well-written and easy to follow, with a comprehensive analysis included.

**Weaknesses:**

- The evaluation could be more thorough by incorporating existing benchmarks such as Codec-Superb and DASB, allowing for a more comprehensive comparison of the proposed method against existing models under standardized settings.

- The related works section could be expanded to include methods that use frequency domain inputs, such as those discussed in the following papers:
    -  https://arxiv.org/pdf/2406.05298
    -  https://arxiv.org/pdf/2201.09429
    -  https://arxiv.org/pdf/2405.00233
    -  https://arxiv.org/pdf/2402.10533
    -  https://www.arxiv.org/pdf/2406.07422
- While Hubert-KM, Encodec, and Speech Tokenizer are reasonable baselines, it would be beneficial to include additional baselines with more similar setups, such as SPECTRAL CODECS (https://arxiv.org/pdf/2406.05298) or SemantiCodec (https://arxiv.org/pdf/2405.00233), for a fuller assessment.
- The proposed model is only evaluated on speech data, leaving other domains, such as general audio and music, unexplored.

**Questions:**

refer to weaknesses

---

> ### Author Response · Authors · 2024-11-17
>
> We thank reviewer for the comments and recognition of our approach. We have thoroughly examined each point and would like to provide detailed responses:
>
> ## 1. Evaluation with Recent Benchmarks
>
> > Evaluate with recent Benchmark such as Codec-Superb and DASB. Compare with more baselines include: SPECTRAL CODECS and SemantiCodec.
>
> We highly appreciate the reviewer for pointing out these relevant work and benchmarks, and we definitely would like to discuss these work in our next revision. However, we believe that these works should be considered as concurrent work, and according to ICLR's concurrent policy [https://iclr.cc/Conferences/2025/FAQ], that paper published within the last 4 months are considered as contemporaneous. Also, paper not published in peer-reviewer proceedings or journals are not required to compare. Therefore, we believe this should not be considered as a main weakness.
>
> ## 2. Evaluation on Other Audio Domains
>
> > Evaluation on other domain audio data includes music and general audio
>
> As indicated in our title and our limitation section, the current scope of this work is primarily on speech. We will consider extending dMel to music and general audio as our future work.

---

### Author Response · Authors · 2024-11-22
**Follow-up on Rebuttal Discussion**

Dear ICLR Reviewers,

I hope this message finds you well. As the discussion period comes to an end, we look forward to receiving your assessment of our rebuttal for paper 5027 and how the clarifications address your initial concerns.

Your considered evaluation of the rebuttal will help ensure a thorough review process. Please let us know if any aspects would benefit from additional clarification from our side.

Thank you for your continued engagement with our work.

Best regards,
Authors of Submission 5027

---

### Author Response · Authors · 2024-11-27
**Rebuttal Revision**

Dear Reviewers,

Thank you for your thorough engagement with our work. We have carefully revised our submission based on your valuable feedback, with all changes highlighted in blue.

Our approach represents a significant departure from conventional methods, which necessitated extensive discussion of several key comparisons, particularly regarding bit-rate metrics and their impact on model complexity. While our method is less dependent on bit-rate considerations, we acknowledge its importance in the literature and have maintained a comprehensive discussion of these metrics to facilitate meaningful comparisons with existing approaches.

We remain grateful for your constructive feedback and are committed to further improving our paper to meet the high standards of the conference.

Best regards,
Authors of Submission 5027

---

### Author Response · Authors · 2024-12-04
**New Experiments Speech Synthesis with ultra-low frame rate dMel feature**

Dear Reviewers,

We appreciate your unanimous agreement that dMel achieves strong results while reducing downstream model complexity. We are excited to share additional results that further strengthen our claims about dMel's efficiency.

Key New Findings:
1. We successfully pushed dMel's frame rate from 40Hz to unprecedented low levels (20Hz, 13.3Hz, and 10Hz) while maintaining competitive performance.
2. At 20Hz, our model achieves 5.0 WER on LibriSpeech test-clean, outperforming USLM's 6.5 WER despite using less than half the frame rate (USLM uses 50Hz).

Technical Details:
Our approach concatenates k frames into k*80 channels while maintaining the same model architecture. This reduces sequence length by factor k:
- k=2: 20 frames/second
- k=3: 13.3 frames/second
- k=4: 10 frames/second

While our submission used k=1 (predicting 80 channels per step), these new experiments use larger k values, where the model predicts k\*80 channels in each step and each speech embedding is derived from k\*80 dMel features.

To our knowledge, we are the first to successfully operate at such low frame rates (10-20Hz) for speech synthesis using just a vanilla Transformer Decoder—a simple, well-investigated architecture! This achievement is unprecedented and addresses a key scalability challenge in speech modeling by significantly reducing sequence lengths.

Complete Results:
|  | Feature | # Frames/second | WER | CER |
| --- | --- | --- | --- | --- |
| VOXTLM (official results) | HuBERT-KM | 50 | - | 3.5 |
| USLM (official results) | SpeechTokenizer | 50 | 6.5 | - |
| RichTTS (our implementation) | HuBERT-KM | 50 | 9.5 | 4.3 |
| RichTTS (our implementation) | SpeechTokenizer | 50 | 11.4 | 5.9 |
| RichardTTS | dMel  | 40 | 4.3 | 1.8 |
| **Our Submission's Results Above / New Low Frame-Rate Results Below** | --- | --- | --- | --- |
| RichardTTS | dMel  | 20 | 5.0 | 2.2 |
| RichardTTS | dMel  | 13.3 | 6.8 | 3.9 |
| RichardTTS | dMel | 10 | 8.2 | 5.0 |

These results further validate our core contributions:
1. dMel's efficiency as an encoder-free, low frame-rate feature
2. The effectiveness of our channel-wise feature encoding/decoding design

We understand that these new results may not be considered in the review process given the timing, but we believe they provide valuable additional validation of our approach.

---

### Meta-Review · Area_Chair_83cy · 2024-12-08

**Metareview:**

This paper proposes dMel, a simple method for quantizing Mel spectrograms into discrete units. Unlike self-supervised semantic tokens and neural codecs, dMel is model-free. The authors train a transformer-based language model for speech-text modelling and evaluate their proposed tokenization approach on ASR and TTS. Experimental results indicate superior ASR and TTS performance compared to prior methods like HuBERT + K-means and SpeechTokenizer. Reviewer qtqg and wcjv requested an evaluation under the framework of spoken LM. The authors noted that some spoken tokenization papers also use similar downstream tasks. Both reviewers qtqg and wcjv suggested that the comparison is unfair due to dMel's higher bit rate. The authors argued that a comparison under the same bit rate is not required, but the reviewers remained not fully convinced. The meta-reviewer supports that a comparison under a similar bit rate is crucial (please refer to Additional Comments on Reviewer Discussion).

**Additional Comments On Reviewer Discussion:**

The paper demonstrates that dMel performs well, providing sufficient evidence in ASR and TTS compared to previous speech tokenization approaches. The main concern, however, is that dMel has a much higher bit rate than other tokens. Whether a comparison of speech tokenization under the same bit rate is necessary remains a point of contention.

Both the reviewers and the AC consider that bitrate should not be ignored. The downstream models and tasks used by the authors are insufficient to justify that bitrate can be overlooked, as it is expected that high-bitrate methods have an advantage under the current setting.


=== Below is the opinion of AC ===

Do we need to consider the bitrate when designing a speech tokenization approach? Let's revisit the rationale for using discrete units or speech representation learning in downstream tasks. The original signal contains more information than its quantized, compressed, or encoded version. If sufficient downstream training data and a sufficiently capable downstream model are available, using the original data outperforms the compressed version, making quantization or representation learning unnecessary. This is also demonstrated in this paper, where continuous mel-spectrograms outperform dMel in both TTS and ASR tasks.

The research on tokenization or compression aims to find better speech representations that allow for less data or smaller downstream models while still achieving strong performance on downstream tasks. From this perspective, the best way to evaluate a speech representation or tokenization approach is to test it under various low-resource settings. However, not all papers evaluate models in this comprehensive manner. Instead, researchers often use bitrate as a proxy. The assumption is that representations with higher bitrates are more complex, and simpler representations are preferred because more complex ones tend to overfit more easily. This assumption is not always accurate, as other factors, such as the amount of data available for downstream tasks, also play a role. This explains why, in general, higher bitrates tend to lead to better performance (see Codec-SUPERB: https://arxiv.org/abs/2402.13071). However, exceptions to this trend have also been noted, as mentioned in the rebuttal.

However, this does not mean that bitrate can be disregarded during comparisons. If we want to disregard bitrate, the authors should verify that the proposed approach performs well in low-resource scenarios (e.g., with less training data or smaller downstream models). Overall, I support the reviewers' view that the consideration of bitrate cannot be ignored.

---

### Decision · Program_Chairs · 2025-01-22

Reject